# Self-aligned patterning of tantalum oxide on Cu/SiO$_2$ through redox-coupled inherently selective atomic layer deposition

Yicheng Li[1,4], Zilian Qi[1,4], Yuxiao Lan[2], Kun Cao[1] ✉, Yanwei Wen[2], Jingming Zhang[2], Eryan Gu[1], Junzhou Long[1,3], Jin Yan[1], Bin Shan [2] & Rong Chen [1,3] ✉

Atomic-scale precision alignment is a bottleneck in the fabrication of next-generation nanoelectronics. In this study, a redox-coupled inherently selective atomic layer deposition (ALD) is introduced to tackle this challenge. The 'reduction-adsorption-oxidation' ALD cycles are designed by adding an in-situ reduction step, effectively inhibiting nucleation on copper. As a result, tantalum oxide exhibits selective deposition on various oxides, with no observable growth on Cu. Furthermore, the self-aligned TaO$_x$ is successfully deposited on Cu/SiO$_2$ nanopatterns, avoiding excessive mushroom growth at the edges or the emergence of undesired nucleation defects within the Cu region. The film thickness on SiO$_2$ exceeds 5 nm with a selectivity of 100%, marking it as one of the highest reported to date. This method offers a streamlined and highly precise self-aligned manufacturing technique, which is advantageous for the future downscaling of integrated circuits.

The semiconductor industry continues to develop smaller and better-performing nano-electronic devices; high-resolution patterning is a critical step in determining the manufacturability of such devices. Misalignment impedes high-precision patterning because nanodevices tend to shrink to the atomic scale[1–6]. Traditional top-down "deposition-lithography-etch" multiple manufacturing steps are limited by significant challenges such as control of edge-placement error (EPE) and complexity of the steps[7]. Moreover, new materials and 3D nanostructures have stringent requirements for high-volume manufacturing (HVM) technology[8–12]. Atomic layer deposition (ALD) is a powerful thin film manufacturing technology, wherein thin films are grown by self-limiting chemical reactions between the precursors and the substrate[13]. Moreover, selective ALD is a promising technique because it allows the atomic-scale precision alignment with simplified steps[14–19]. Selective ALD enables deposition of films only on the desired regions of pre-patterned substrates. This method promotes effective

nanoelectronics manufacturing. For example, previous studies have adopted a selective dielectric-on-dielectric ALD process to create a dielectric scaffold that prevented the vias from getting too close to the neighboring metal features, leading to a more significant process margin for EPE during via formation[20,21].

Selective ALD between dielectric oxides and metals was achieved through surface passivation. Polymers[22], self-assembled monolayers (SAMs)[23–27], and small molecule inhibitors[28,29] have been used to block nucleation in non-growth areas. For example, SAMs have been used to block nucleation on Cu/Co regions to achieve fully self-aligned via (FSAV) integration. This approach demonstrated improvement of two orders of magnitude in the via-to-line time-dependent dielectric breakdown for interconnect scaling beyond the 3 nm node[30]. Moreover, the selective tungsten filling technology eliminated MOL/BEOL parasitic resistance caused by the liner/barrier and seed layer, resulting in reduced resistivity and improved circuit performance[31]. These

---

[1]State Key Laboratory of Intelligent Manufacturing Equipment and Technology, School of Mechanical Science and Engineering, Huazhong University of Science and Technology, Wuhan, Hubei, People's Republic of China. [2]State Key Laboratory of Materials Processing and Die & Mould Technology, School of Materials Science and Engineering, Huazhong University of Science and Technology, Wuhan, Hubei, People's Republic of China. [3]Hubei Yangtze Memory Laboratories, Wuhan, Hubei, People's Republic of China. [4]These authors contributed equally: Yicheng Li, Zilian Qi. ✉e-mail: kuncao@hust.edu.cn; rongchen@mail.hust.edu.cn

inhibitors assisted selective ALD methods are close to practical industrial applications, but challenges still exist. This process requires a long immersion time and subsequent removal of SAMs or inhibitors. Moreover, severe limitations exist because the blocking effect of SAMs deteriorates during the deposition process and is heavily temperature-dependent[32]. Other inhibitors suffer from limited materials and process parameters required for reliable selectivity[33–35]. Thus, as the critical size decreases below 10 nm and the demand for new metallic and dielectric materials and 3D nanostructures continue to increase, the appropriate chemicals for passivation-assisted selective ALD become very important. Inherently selective ALD is a more straightforward method for alignment manufacturing, that is free of inhibitor passivation and removal steps[36–39]. In particular, ALD relies on intrinsic surface properties differences. For example, inherently selective deposition process has been reported for oxides deposited on noble metals such as Ru, Pt. The selectivity originates from the catalytic combustion of the precursor ligands and preferential dissociation of co-reactants on metals. Thus, the target film is deposited on the metal, but not the dielectric[40,41]. In contrast, selective deposition of dielectric on dielectric is challenging to suppress nucleation on metals while ensuring deposition on dielectrics. This process is vital for FSAV fabrication becasue it increases the spacing between vias and metal lines. Hence, the use of reductive co-reactants to reduce metal catalytic activity is a promising method. Some chemicals, such as acetic acid[42], ethanol[43–45], isopropyl alcohol[46], and tert-butylamine[47], are used to prevent surface oxidation and undesired nucleation on the metal. Nonetheless, nucleation on metal is still difficult to suppress[48]. Inherently selective ALD is very sensitive to surface chemistry, and adjustment of the metal surface is an essential factor for selective ALD.

Herein, a redox-coupled inherently selective ALD is developed that effectively inhibits nucleation on Cu. Tantalum oxide exhibits selective deposition on various oxides, with no observable growth on copper. $TaO_x$ films are widely used as insulating layer for nanoelectronics, functional layer for memory devices, etc[49–52]. This study reveals that the loss of selectivity is attributed to the surface oxidation of Cu, and proposes the "reduction-adsorption-oxidation" ALD cycles. The optimized selectivity of 100% is achieved, and the maximum thickness of the film deposited on $SiO_2$ is 5–6 nm. This work is one of the highest reported to date. During the redox-coupled ALD process, the EtOH pulse before each binary ALD cycle can reduce surface oxidation in situ and suppress undesired nucleation on Cu. The selectivity originates from the higher energy barrier during ALD nucleation on reduced Cu than that of OH-terminated $SiO_2$. Finally, the selective deposition approach is transferred for self-alignment on nanoscale $Cu/SiO_2$ patterns, and excessive 'mushroom' growth at the edges and formation of nucleation defects on the Cu region are not observed. The ALD method provides a streamlined bottom-up avenue for self-alignment nanomanufacturing and unfolds possibilities for semiconductor applications.

## Results and discussion

### AB and ABC-type ALD processes

Traditional binary ALD was optimized by incorporating an in-situ reduction step into each binary AB-type ALD cycle. The 'reduction-adsorption-oxidation' ALD cycles were designed, and the scheme of this redox-coupled ABC-type (co-reactant A → precursor B → co-reactant C) ALD is presented in Fig. 1a. An additional reduction pulse continuously mitigated the oxidation of Cu regions, which further improved the selectivity between Cu and oxides (including $SiO_2$, $Al_2O_3$, and $HfO_2$). The in-situ reduction step has potential to be integrated into the industrial process without the requirement of long-term liquid passivation with SAMs and removal steps.

The deposition temperature is essential for selective ALD. First, the AB-type ALD process with $Ta(N^tBu)(NEt_2)_3$-$H_2O$ as precursors was conducted at 100, 200, and 300 °C. When the deposition temperature

was decreased to 100 °C (Supplementary Fig. 1), the average deposition rate on $SiO_2$ and Cu is 0.16 and 0.14 nm/cycle, respectively. The selectivity of the AB-type ALD process decreased significantly, which was probably attributed to the partial condensation of the precursor on the substrate. When the temperature was increased to 300 °C (Supplementary Fig. 2), the selectivity between Cu and $SiO_2$ also deteriorated. This may be ascribed to the oxidation of Cu and the decomposition of the precursors[53]. At 200 °C (Supplementary Fig. 3), ALD initial growth rate on Cu was inhibited by using EtOH and $H_2O$ as co-reactant. The selectivity is quantified with the formula $(\theta_{GA}-\theta_{NGA})/(\theta_{GA} + \theta_{NGA})$, $\theta_{GA}$ is the thickness or amount of material deposited on the growth region, and $\theta_{NGA}$ is the thickness or amount of material on the non-growth. The selectivity of ALD process performed with $Ta(NtBu)(NEt_2)_3$-$H_2O$ was obtained 32%, 91%, and 88% at 100 °C, 200 °C, and 300 °C, respectively (Supplementary Fig. 4). The saturate time of Ta precursor on $SiO_2$ was ~2 s. Decreasing the Ta precursor pulse time to 1 s, the nucleation delay on Cu could be maintained to 100 cycles while the growth rate on $SiO_2$ was too slow. Increasing the pulse time to 3 s, the selectivity deteriorated due to quick nucleation on Cu. (Supplementary Fig. 7).

Then, the redox-coupled ABC-type ALD performed with EtOH-$Ta(N^tBu)(NEt_2)_3$-$H_2O$ as precursors at 200 °C was studied. An apparent nucleation delay of more than 150 cycles on the Cu surface was observed (Fig. 1b). It was found that the film thickness could be lower than zero when ethanol was utilized in the ALD process. This was reasonable as ethanol could reduce the native oxide of copper. To study it further, bare copper substrates were tested using SE measurements after exposure to ethanol, $H_2O$, $O_3$ pulses (Supplementary Fig. 8). The results showed that the decrease of surface oxide layer thickness with ethanol pretreatment was about 0.5 nm. $H_2O$ had minimal influence on the surface oxide layer, while $O_3$ strongly oxidized the Cu surface, thereby increasing the surface oxide layer. The surface reduction process was also reported capable to improve the interface and film quality[54]. The surface morphology of the $TaO_x$ films deposited on $SiO_2$ was studied through atomic force microscopy (AFM) (insets in Fig. 1b). The roughness of the original substrates was evaluated through AFM in Supplementary Fig. 5. After the ALD process, the surface roughness values were 0.50, 0.45, and 0.48 nm after 50, 100, and 150 ABC-type ALD cycles, respectively. Barely particles were observed on the $SiO_2$ and Cu surface (Supplementary Figs. 5 and 6), indicating that smooth film was obtained. For the ABC-type ALD, it was found that the growth rate of $Ta_2O_5$ on $Al_2O_3$ and $HfO_2$ was lower than that on $SiO_2$, which confirmed the occurrence of previously reported surface acidity-induced selective deposition[55]. A total of 10 s purge time was sufficient to remove excess precursors and by-products. High-precision X-ray photoelectron spectroscopy (XPS) was conducted to quantitatively compare the amount of $TaO_x$ on Cu and $SiO_2$. At 50 and 100 ALD cycles, peak ascribed to Ta 4 f was not observed, indicating barely growth of $TaO_x$ on Cu (inset in Fig. 1c). The proportion of Ta was almost zero on the Cu surface at 50 cycles and 100 cycles indicating 100% selectivity. With increasing number of ALD cycles to 200, the peak intensity of Ta on the $SiO_2$ and Cu substrates increased simultaneously. The Ta element ratios were 39% for Cu and 77% for $SiO_2$ after 200 ALD cycles, resulting in loss of selectivity (Fig. 1c).

A nucleation model proposed by Parsons[18] is adopted and varied to fit the selective ALD process, as indicated by the line curves in Fig. 1b. The nucleation model includes the factors of the normal ALD nucleation ($\dot{N}$ ($nm^{-2}$)), defect induced nucleation ($\hat{N}$ ($nm^{-2}$)), the anisotropic growth of the existed nucleus, and atomic diffusion induced nucleation in the dynamic expanding region at the edge of nucleus ($\dot{N}'$ ($nm^{-2}$)). The details of the nucleation model are described in Supplementary Materials (Supplementary Fig. 9). The fitting curves are consistent with the experimental results, and the proper error $f_{err}$ is less than $2 \times 10^{-2}$. For ABC-type ALD processes on $SiO_2$ and $Al_2O_3$

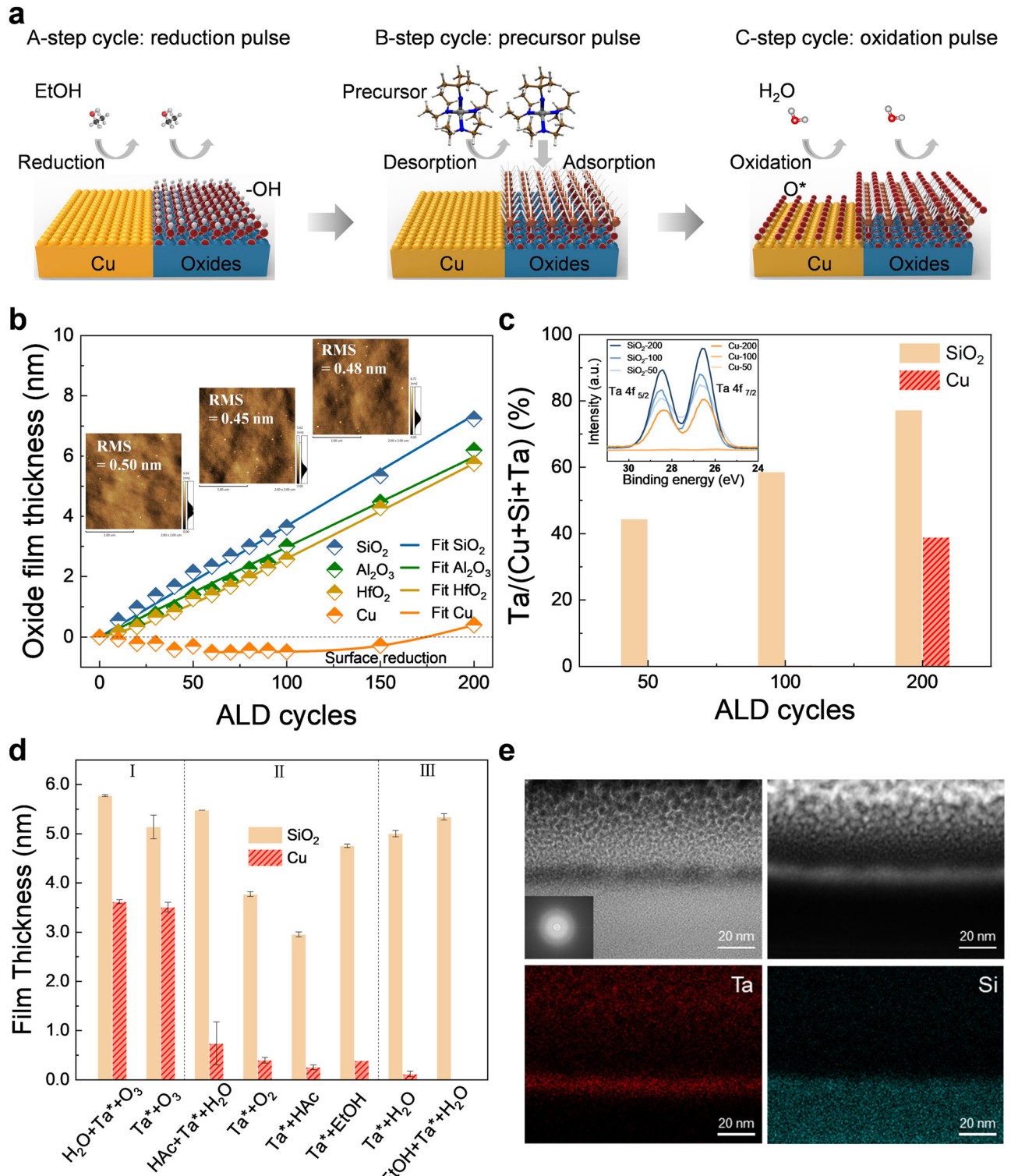

**Fig. 1 | The selective ALD processes. a** The scheme of redox-coupled ABC-type ALD. **b** The TaO$_x$ film thickness as a function of ALD cycles, the line curves are fitting data through the nucleation model, and the insets show the AFM images and the root-mean-square (RMS) roughness of the films corresponding to 50, 100, and 150 ALD cycles. **c** The proportion of Ta element on Cu and SiO$_2$ during 50, 100, and 200 ALD cycles, respectively, the inset shows the corresponding high-resolution XPS scan of Ta 4 f. **d** Film thickness versus different AB-type and ABC-type ALD processes between Cu and SiO$_2$. Each data point of the film thickness is measured at two positions of one sample by spectroscopic ellipsometry. Error bars represent standard deviations after two measurements of each sample. **e** The cross-sectional bright-field and dark-field TEM images, and element scans of Ta and Si of the TaO$_x$ thin film, the insetting electron diffraction image shows the film is amorphous.

substrates, the values of nucleation site density induced per ALD cycle on non-defect sites $\dot{N}$, defect-induced nucleation site density $\hat{N}$ are at the order of $10^{-1}$ nm$^{-2}$. This suggests that the rapid and linear growth observed on these surfaces is associated with the formation of a significant number of nucleation sites. The $G_v$ value of TaO$_x$ on Al$_2$O$_3$ is fitted to be about 62% that of SiO$_2$ in Supplementary Fig. 10, which agrees with the experimental ratio 78%. (Supplementary Table 1) The values of $\dot{N}$ and $\dot{N}'$ on the HfO$_2$ substrate are on the order of $10^{-3}$ nm$^{-2}$,

which is significantly lower than the values on the $SiO_2$ and $Al_2O_3$ substrates, resulting in slow initial nucleation rate. The values of $\dot{N}$ and $\hat{N}$ on Cu are $2.9 \times 10^{-7}\,nm^{-2}$ and $1.1 \times 10^{-4}\,nm^{-2}$, respectively, which are much lower than those on oxide substrates (Supplementary Fig. 10). The results suggest that defect-induced nucleation is limited on Cu, and those parameters $\dot{N}$ and $\hat{N}$ are the critical factors for nucleation delay.

For selective ALD process optimization, a series of AB- and ABC-type ALD processes were conducted and compared in Fig. 1d. The reductant (EtOH), acid (HAc), and oxidants ($H_2O$, $O_2$, and $O_3$) were studied as co-reactants. For the AB-type ALD, the growth rate on $SiO_2$ with different co-reactants decrease as follows: $H_2O > O_3 > O_2 > HAc > EtOH$ (Supplementary Fig. 11). $H_2O$ provides more active hydroxyl sites on the surface and promotes the precursor adsorption. It is expected that the next half-reaction of ALD may be different for various co-reactants[56]. Except for the $O_3$ process, an apparent nucleation delay on the Cu surface is observed. (Supplementary Fig. 3) The $O_3$ co-reactant strongly oxidizes the Cu surface and offers active surface sites to initiate growth; thus, there's no nucleation delay on the Cu surface. The EtOH as a co-reactant could achieve high selectivity; however, it is difficult to oxidize the Ta precursor; thus, the deposition rate is slow (Supplementary Fig. 3a, b). The film thickness is <5 nm after 650 ALD cycles, which is time-consuming and precursor-wasting. Although the $H_2O$-based ALD process has high selectivity, the nucleation delay is quickly lost after 50 ALD cycles. Other Ta precursors with different coordinating groups were also exploited, including $Ta(NMe_2)_5$ and $Ta(OEt)_5$. However, both precursors need highly-active $O_3$ as co-reactant, thus the Cu substrate is strongly oxidized which is harmful to the selective ALD process (Supplementary Fig. 12). Different ABC-type ALD processes (co-reactant A → precursor B → co-reactant C) are developed, including EtOH-$Ta(N^tBu)(NEt_2)_3$-$H_2O$, HAc-$Ta(N^tBu)(NEt_2)_3$-$H_2O$, $H_2O$-$Ta(N^tBu)(NEt_2)_3$-$O_3$. The EtOH-$Ta(N^tBu)(NEt_2)_3$-$H_2O$-type process exhibits the highest selectivity of 100% and can be maintained for 100 cycles (Supplementary Figs. 13, 14). During each ALD cycle, the co-reactant in the A-step is proposed to be utilized for in situ surface reduction, followed by two subsequent half-reactions, including the chemisorption of the precursor and the subsequent oxidation reaction with the co-reactants. To suppress nucleation in the non-growth Cu region, the EtOH-based A-step is preferred over other ABC-type processes. For the HAc-$Ta(N^tBu)(NEt_2)_3$-$H_2O$ ABC-type ALD process, acetic acid slightly etches Cu surface; and it provides more active sites for ALD nucleation that deteriorated the selectivity between Cu and $SiO_2$.

The selective ALD processes of $TaO_x$ on Cu/$SiO_2$ could be divided into three groups (Fig. 1d). The $O_3$-based process in region I is not appropriate for selective deposition, as it strongly oxidized the Cu surface. In region II, the EtOH-based AB-type binary process is also unsuitable because of its low growth rate. The selectivity of the $H_2O$-based AB-type binary process is higher than that of the highly active $O_3$ at the initial growth stage within 50 cycles, but decreases quickly. In region III, the EtOH-$Ta(N^tBu)(NEt_2)_3$-$H_2O$ ABC-type process is the most preferred one. It achieves the highest selectivity and the longest nucleation delay on the non-growth Cu region, both are critical for a reliable self-aligned oxide stacking. Nucleation on Cu is still inhibited when the film thickness is obtained ~5–6 nm on $SiO_2$. Figure 1e shows the cross-section TEM images of $TaO_x$ film on $SiO_2$ after 150 ABC-type ALD cycles; the film is continuous and amorphous. GI-XRD results confirm the amorphous structure (Supplementary Fig. 15). For inherently selective ALD, there is a trade-off between selectivity and film thickness on $SiO_2$. In this work, the selectivity between Cu and $SiO_2$ is compared with those reported in previous studies. The obtained selectivity (100%) is the highest among those reported for other developed inherently selective ALD strategies. (Supplementary Fig. 16).

## Investigations of Cu surface chemical state to selective ALD

The nucleation delay is influenced by Cu surface chemical state. To investigate the origin of the selectivity loss, Cu surface with different pretreatments followed by AB-type ALD ($Ta(N^tBu)(NEt_2)_3$-$H_2O$) process are conducted, including EtOH-treated Cu, HAc-treated Cu, $O_2$-treated, and $O_3$-treated Cu (Fig. 2a). EtOH-reduced Cu exhibits a high selectivity (91.2% during 50 ALD cycles). With HAc immersion pre-treatment, the thickness of the Cu film decreases, indicating that the surface Cu oxides is slightly etched. HAc-etched Cu exhibits high selectivity (76.8% during 50 ALD cycles). With the oxidizing atmosphere treatment, the thickness of the Cu films increased, indicating that the surface is strongly oxidized to CuO. The selectivity for $O_2$-oxidized and $O_3$-oxidized Cu is decreased to 37.8% and 8.4%, respectively. For the $O_2$-treated Cu surface, nucleation occurs quickly, and the growth rate is similar to that of the $SiO_2$ substrate after 20 ALD cycles, indicating that surface oxidation decreases selectivity. The $O_3$-treated Cu surface exhibits linear growth behavior similar to that of $SiO_2$, which is also attributed to oxidation and activation with ozone. The growth curves on different Cu surfaces are fitted with the nucleation model, the key parameters are summarized in Supplementary Table 2. All fitting errors ($f_{err}$) are less than $4 \times 10^{-2}$. For EtOH-treated Cu, $\dot{N}$ is the lowest; thus, the nucleation delay is the longest. $\dot{N}$ and $\dot{N}'$ increase roughly in the order of EtOH-treated Cu, HAc-treated Cu, $O_2$-treated Cu, and $O_3$-treated Cu. Thus, nucleation delays decreased in the same order.

The comparison of selectivity influenced by different pretreatments followed with AB-type ALD ($Ta(N^tBu)(NEt_2)_3$-$H_2O$) process is summarized in Fig. 2b. The oxidation of Cu with $O_2$ and $O_3$ is unfavorable for initiating nucleation on Cu. The Cu $2p$ XPS spectra are obtained (Fig. 2c). In the spectra, $Cu^{2+}$ satellite peaks were observed for oxidized Cu, and the peaks located at 933.8 and 935.4 eV are ascribed to $Cu^0/Cu^{1+}$ and $Cu^{2+}$, respectively (Supplementary Fig. 17). Except for oxidized Cu, other samples show a low-intensity $Cu^{2+}$ satellite peak. The oxidation ratio of Cu during ABC-type ALD process is also presented in Fig. 2d. The original Cu surface has $Cu^{2+}$ proportion of 48%. After 50 cycles with ABC-type ALD, the $Cu^{2+}$ concentration decreases to 25%, indicating reduction of the Cu surface during EtOH pulses. As the number of ALD cycles increase, the Cu oxidation state slightly increase to 30% for 100 ALD cycles and 33% for 200 ALD cycles. In contrast, the ratio of $Cu^{2+}$ could be decreased to zero after only EtOH pulses treatment, indicating the reduction of Cu by EtOH is very effective. Thus, during ABC-type ALD, the reduction during the EtOH pulse and oxidation during the Ta precursor and $H_2O$ pulses competitively occur on Cu films. Oxidation ultimately dominated and led to the nucleation of precursors on Cu.

The reduction of Cu oxides by EtOH is presented:

$$CuO + CH_3CH_2OH \rightarrow Cu_2O + H_2O(g) + CH_3CHO(g) \tag{1}$$

$$Cu_2O + CH_3CH_2OH \rightarrow Cu + H_2O(g) + CH_3CHO(g) \tag{2}$$

The etching of Cu oxides by HAc is presented:

$$CuO + CH_3COOH \rightarrow Cu(CH_3COO)_2 + H_2O(g) \tag{3}$$

## The surface adsorption and reaction products analysis

An in-situ quartz crystal microbalance (QCM) was used to investigate the nucleation behavior. According to the Sauerbrey equation, the attenuation of the resonant frequency of the crystal oscillator was proportional to the minor mass change during deposition[57]. The mass gain during a single ALD cycle was presented (Fig. 3a). On the $SiO_2$ surface, it was found to be much higher than that on Cu, according to the data in the insets of Fig. 3a. The mass gain on Cu and $SiO_2$ during

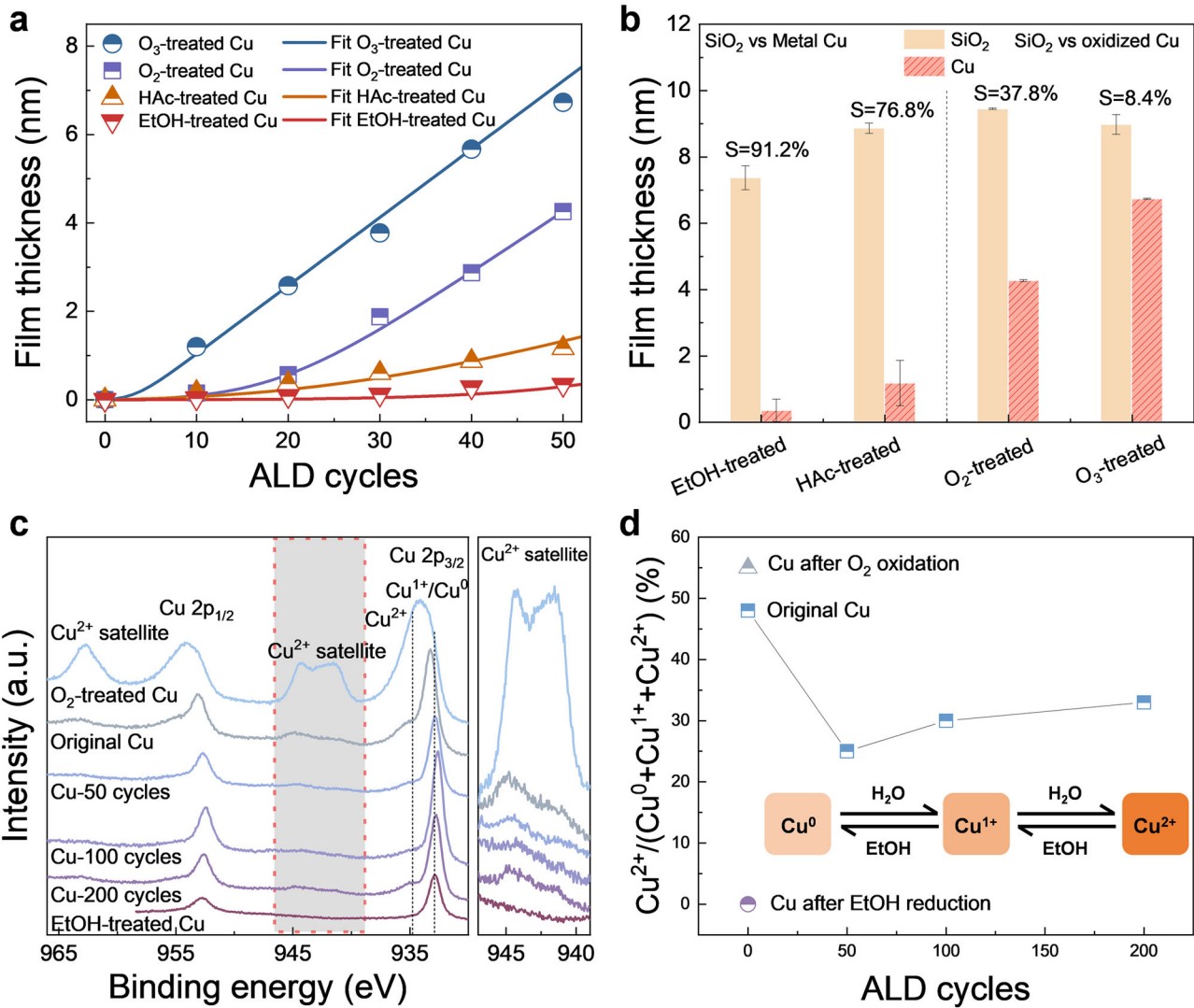

**Fig. 2 | The effect of chemical state of Cu surface to selective ALD. a** The film thickness *versus* ALD cycles for O$_3$-treated, O$_2$-treated, HAc-treated, and EtOH-treated Cu. **b** The film thickness and selectivity as a function of the ALD cycles on SiO$_2$ and Cu substrates with different treatments. The selectivity is quickly lost for oxidized Cu. Each data point of the film thickness is measured at two positions of one sample by spectroscopic ellipsometry. Error bars represent standard deviations after two measurements of each sample. **c** The Cu *2p* scan of XPS and **d** the proportion of Cu$^{2+}$ after O$_2$ oxidation, EtOH reduction, and during 0, 50, 100, and 200 ALD cycles. The highlighted region on the left of **c** means the Cu$^{2+}$ satellite of Cu *2p*, which is enlarged as the right of **c**. The insets in **d** show the competition between the removal of Cu oxides during EtOH dosing and the oxidation of the Cu surface during H$_2$O dosing.

AB-type (Ta(N$^t$Bu)(NEt$_2$)$_3$-H$_2$O) ALD at (a) 100 °C and (b) 300 °C are also shown in Supplementary Fig. 18. The average mass gains of SiO$_2$ and Cu after each precursor's pulse are presented in Fig. 3b. Increasing the growth temperature promotes the desorption of ALD precursors; thus, the mass gain measured by the QCM decreases, which is consistent with the results of the ellipsometry tests. The adsorption of water is also influenced with deposition temperature, more H$_2$O molecules are physically adsorbed at lower temperature, thus the initial mass gain is larger after dosing H$_2$O at 100 °C. During ABC-type ALD, the adsorbed EtOH is found to hinder the subsequent adsorption of Ta precursors on SiO$_2$ in Fig. 3c. EtOH is also capable to convert surface hydroxyl groups to ethoxide groups and act as an inhibitor to block adsorption sites. This is another reason to suppress nucleation on Cu surfaces (Supplementary Fig. 19). It should be noted that the growth rate on Cu surface with optimized ABC-type ALD is not suppressed to zero by QCM measurements, which may be a result of a highly rough Cu morphology deposited on crystal oscillator (Supplementary Fig. 20). For AB-type ALD reactions, HNEt$_2$ and [HNEt$_2$-CH$_3$]$^+$ can be detected with in-situ quadrupole mass spectrometer

when the Ta precursors are dosed (Supplementary Fig. 21). The results indicate that the H-transfer reaction occurs between the hydroxyl groups and Ta precursors on SiO$_2$. In addition, the partial pressure of the by-products during the precursor pulse in the ABC-type process is lower than that in the AB-type process, indicating that the EtOH pulses may hinder the chemisorption of the Ta precursor. To compare the composition and dielectric constant of TaO$_x$ films fabricated by AB-type (Ta(N$^t$Bu)(NEt$_2$)$_3$-H$_2$O) and ABC-type (EtOH-Ta(N$^t$Bu)(NEt$_2$)$_3$-H$_2$O) ALD, XPS sputter depth profiles are analyzed. The results show that the ethanol used in the ABC-type ALD process do not influence the carbon concentration in the deposited film (Supplementary Fig. 22). The dielectric constants of the tantalum oxide film for AB and ABC ALD processes are 21.6 and 20.6, respectively. The k values are obtained through CV test and the values are similar (Supplementary Fig. 23).

## DFT simulations

To study the origin of the selectivity between Cu and SiO$_2$, DFT calculations are employed. (Table S3, Fig. 3d) The Ta(N$^t$Bu)(NEt$_2$)$_3$

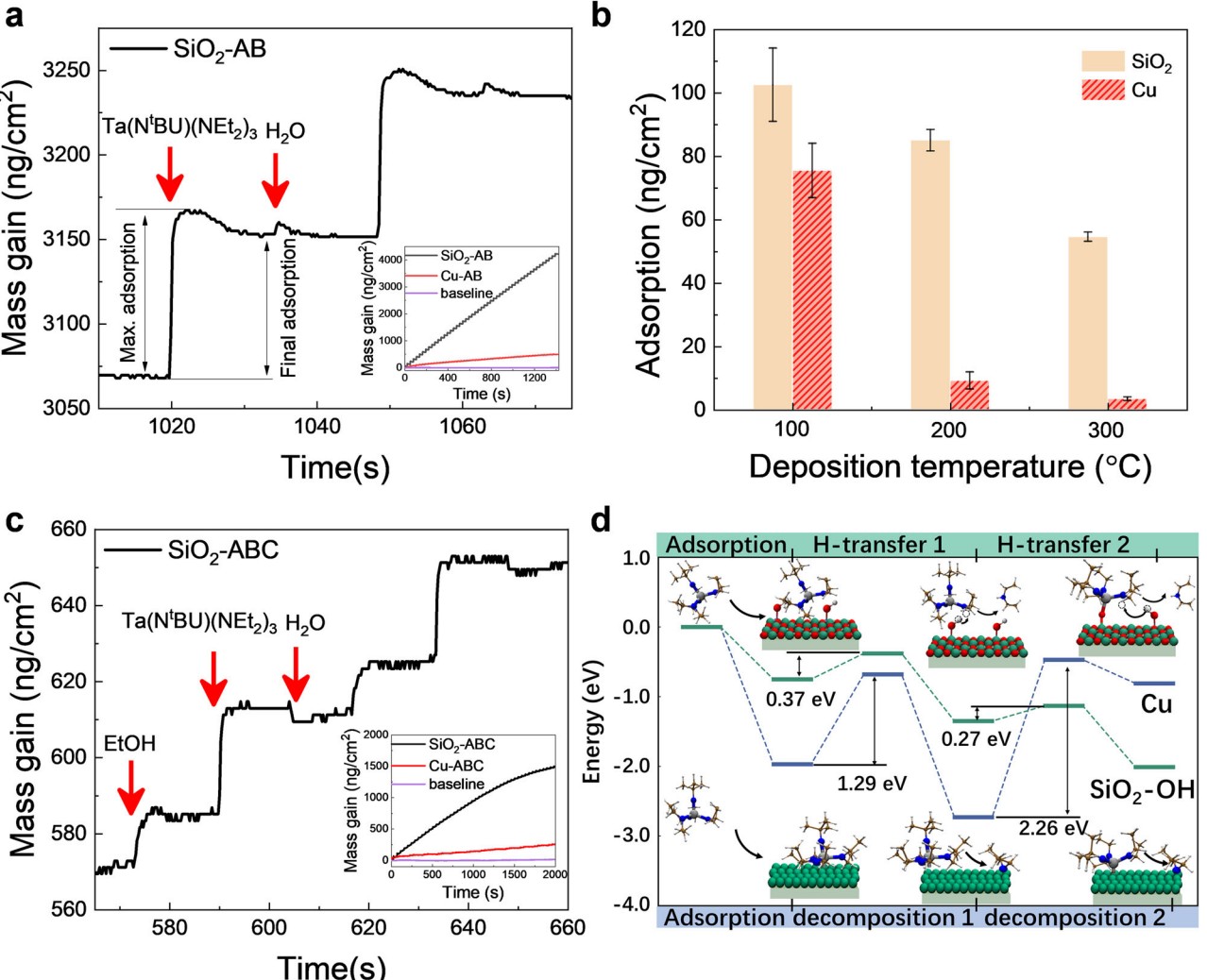

**Fig. 3 | The surface adsorption characterizations of precursors on Cu and SiO2.** **a** The mass gain as a function of deposition time during an AB-type (Ta(N$^t$Bu)(NEt$_2$)$_3$-H$_2$O) ALD cycle, the inset shows the total mass gain on SiO$_2$ and Cu. **b** The mass gain during precursors' pulse at the different growth temperatures. Error bars represent standard deviation of at least three measurements. **c** The mass gain as a function of deposition time during ABC-type (EtOH-Ta(N$^t$Bu)(NEt$_2$)$_3$-H$_2$O) ALD, the inset shows the total mass gain on SiO$_2$ and Cu. **d** DFT calculations of reaction path for reduced Cu and OH-terminated SiO$_2$.

precursor is thought to decompose the -NEt$_2$ ligands on the reduced Cu surface. The reaction equation is shown as follows, where * indicates empty sites, Ta(N$^t$Bu)(NEt$_2$)$_{x-1}^*$ is the remaining ligand on the surface, and NEt$_2^*$ is the ligand decomposes to the empty site:

$$Ta(N^tBu)(NEt_2)_x^* + ^* \rightarrow Ta(N^tBu)(NEt_2)_{x-1}^* + NEt_2^*(x \geq 1) \quad (4)$$

The decomposition of the first NEt$_2^*$ ligand is difficult because of its high reaction barrier ($E_b = 1.29$ eV). In addition, the second -NEt$_2$ decomposition is hindered, as verified by thermodynamics ($\Delta H = 1.92$ eV) and kinetically ($E_b = 2.26$ eV). On the OH-terminated SiO$_2$ surface, H-transfer from the hydroxyl on the surface to the Ta(N$^t$Bu)(NEt$_2$)$_3$ precursor occurrs. The H-transfer reaction is revealed to be a key factor for the inherently selective ALD on oxide substrates[55]. When the precursor reacts on the surface, it is assumed that some -NEt$_2$ and -N$^t$Bu groups are released through ligand exchange reactions with the OH surface groups. The ratio between the two possible reaction by-products, HNEt$_2$(g) and H$_2$N$^t$Bu(g), are reported to be 1.7:0.3[58]. Then, a subsequent H$_2$O pulse transferred the remaining ligands and transformed the surface back to a hydroxyl. The reaction equations are as follows:

H-transfer-1:

$$Ta(N^tBu)(NEt_2)_3^* + OH^* \rightarrow Ta(N^tBu)(NEt_2)_2^* \sim O + HNEt_2(g) \quad (5)$$

$$Ta(N^tBu)(NEt_2)_3^* + OH^* \rightarrow Ta(NEt_2)_3^* \sim O \sim HN^tBu \quad (6)$$

H-transfer-2:

$$Ta(N^tBu)(NEt_2)_2^* \sim O + OH^* \rightarrow Ta(N^tBu)(NEt_2)^* \sim O \sim O + HNEt_2(g) \quad (7)$$

Although two reactions for H-transfer-1 are both possible, the proton is preferentially transferred to the -NEt$_2$ ligands than the -N$^t$Bu ligand on the SiO$_2$ surface. Therefore, hydrogenation of the -NEt$_2$ ligands is mainly considered in the following H-transfer reaction. H-transfer reactions are exothermic with a low reaction barrier on OH-terminated SiO$_2$, indicating that reactions could occur easily on OH-terminated surfaces. Overall, the Ta(N$^t$Bu)(NEt$_2$)$_3$ precursor shows lower reactivity on reduced Cu than that on OH-terminated SiO$_2$ because of the higher energy barrier, which is the origin of the selectivity between the reduced Cu and OH-terminated SiO$_2$.

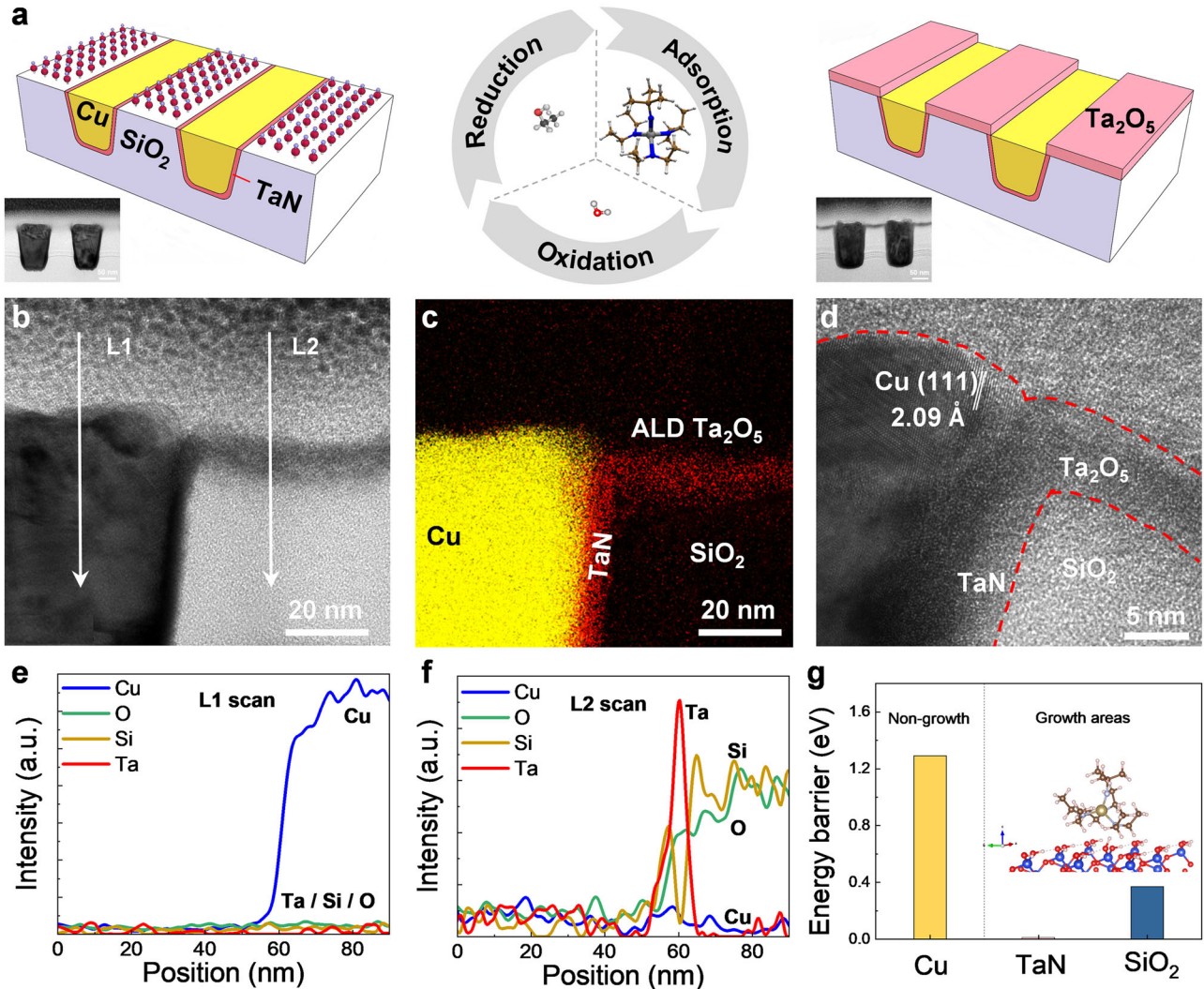

**Fig. 4 | The characterizations of self-aligned TaO$_x$ deposited on Cu/SiO$_2$ nanopatterns. a** The scheme and TEM image of original 50 nm critical dimension Cu/SiO$_2$ patterns and self-aligned patterning of TaO$_x$ film after redox-coupled inherently selective ALD. **b** The cross-section TEM image of self-aligned TaO$_x$ film on the Cu/SiO$_2$ pattern. The L1 and L2 mean the region 1 and 2 for line scan of Cu, O, Si, and Ta elements, respectively. **c** The element mapping of Cu and Ta. **d** The high-resolution TEM image of the border between Cu and SiO$_2$. The line scan across the **e** Cu and **f** SiO$_2$ region after 100 ALD cycles, respectively. **g** The DFT calculation of TaO$_x$ deposited on Cu, TaN, and SiO$_2$, the nucleation on the Cu region is inhibited.

## Selective ALD on Cu/SiO$_2$ nanopatterns

To directly observe the selective deposition on the nanopattern structures, the cross-section TEM of TaO$_x$ deposited on the Cu/SiO$_2$ nanopatterns in a dense via-chain array is performed. Cu/SiO$_2$ patterns were prepared from the chip production line. Schemes and TEM images of the original Cu/SiO$_2$ patterns and self-aligned patterned Ta$_2$O$_5$ films prepared by redox-coupled inherently selective ALD are presented in Fig. 4a. The pitch of the Cu/SiO$_2$ patterns is ~100 nm, and the critical dimension is ~50 nm. After ABC-type ALD, the SiO$_2$ surface is coated with conformal Ta$_2$O$_5$ film, whereas the Cu region remains in its original morphology without any Ta$_2$O$_5$ deposition (Fig. 4b). Through EDS mapping, Ta is detected on the Si region while no signals on Cu regions (Fig. 4c). The element mappings of Si, Cu, and Ta are presented in Supplementary Fig. 24. Enlarged high-resolution TEM image of the border between the Cu and SiO$_2$ is presented in Fig. 4d. The Cu feature is non-oxidized, and the crystallographic orientation is (111). In the enlarged images, there is no mushroom growth at the edge, and no defects are observed within the Cu regions. Element line scans across the Cu and SiO$_2$ regions are presented in Fig. 4e, f. There is no Ta element detected on copper region, while a strong Ta signal is detected on the SiO$_2$, which also confirm the 100% selectivity is achieved.

DFT calculations show that the chemisorption of the Ta(N$^t$Bu)(NEt$_2$)$_3$ precursor on the TaN barrier and SiO$_2$ is thermodynamically favored (Fig. 4g). The reaction barrier on Cu is much larger. Thus, the Ta$_2$O$_5$ film is preferentially deposited on SiO$_2$ and TaN instead of Cu, forming self-aligned patterning. High resolution TEM images of different regions also verified that TaO$_x$ deposition is restricted to the TaN barrier and SiO$_2$, and diffusion of Ta atoms to neighboring Cu regions is unlikely to occur, avoiding excessive mushroom growth at the edges or the emergence of undesired nucleation defects within the Cu region (Supplementary Fig. 25). Several oxides, including NbO$_x$, MoO$_x$, and WO$_x$, have also been tried using AB and ABC-type ALD processes. The results indicated that the redox-coupled ALD method is capable to achieve higher selectivity compared to the binary ALD process, demonstrating the generality of this method (Supplementary Figs. 26–28).

In conclusion, TaO$_x$ is selectively deposited on SiO$_2$ before Cu for self-alignment process. The Ta(N$^t$Bu)(NEt$_2$)$_3$ precursor has a higher energy barrier on reduced Cu surface than that of OH-terminated SiO$_2$ surface, which is the origin of the selectivity. The selectivity decreases with the oxidation of Cu during ALD. Optimized selectivity is achieved with redox-coupled ABC-type (EtOH-Ta(N$^t$Bu)(NEt$_2$)$_3$-H$_2$O) ALD

process, and EtOH is used to reduce the surface oxidation of Cu in situ. Moreover, the inherently selective ALD process is successfully transferred onto $Cu/SiO_2$ nanopatterns with ~100 nm pitch and obtained high selectivity with 5–6 nm films on $SiO_2$ and no defects in the Cu region. The results indicate that inherently selective ALD is a robust and general tool that has excellent application prospects in back-end-of-line processes, which provides an innovative avenue for self-aligned nanostructures.

## Methods

### Substrate preparation

The Si wafer with ~2 nm native oxides were used as the initial $SiO_2$ substrate. The Cu films were evaporated on Si wafer with thickness of ~5 nm. The substrates were stored in an argon atmosphere until ALD process were performed. The quartz crystal oscillators were coated with ~5 nm Cu or $SiO_2$ for quartz crystal microbalance (QCM) tests (Inficon, SQM-160).

### ALD process

The ALD reactions were performed in a custom-built hot-walled ALD reactor (Material Design and Nano-manufacturing center @ HUST, Wuhan, China). $Ta_2O_5$ was deposited using $Ta(N^tBU)(NEt_2)_3$ precursor (>99.9%, Aimou Yuan, Nanjing, China) and various co-reactants, including $O_3$, $O_2$, $H_2O$, EtOH, $CH_3COOH$. All the chemicals are high purity >99.9%. Ozone was prepared with high-purity oxygen (99.999%) through an ozone generator (11 vol% of $O_3$ in $O_2$). The heating temperature of the Ta precursor steel bubbler was 65 °C. The high-purity Ar gas (40 sccm, 99.999%) was injected into the bubbler and carried precursors dosed into the ALD cavity. The pipeline temperature was up to 90 °C to prevent the precursors' condensation.

### Characterization methods

The film thickness was measured by a spectroscopic ellipsometer (M-200X, J. A. Woollam Co.). The modified $TaO_x$ Cauchy model was used to fit the ellipsometer data through the Complete EASE software. J.A.Woollam M2000 spectroscopic ellipsometry was utilized to collect data. Each data point of the film thickness is measured at two positions of one sample by spectroscopic ellipsometry. Error bars represent standard deviations after two measurements of each sample. The surface composition was detected by X-ray photoelectron spectroscopy (XPS, AXIS-ULTRA DLD-600 W). The surface morphology was determined by atomic force microscopy (AFM, SPM9700). AFM was performed in tapping mode using a Molecular Imaging PicoScan Controller. Aluminum reflex coated Si AFM probes were used. Data were processed and analyzed using Gwyddion 2.49 software. The cross-section films were analyzed by transmission electron microscopy (TEM, Talos F200X) with bright-field and annular dark-field scanning modes, and the element distribution was analyzed by energy dispersive X-ray spectroscopy (EDS). The electrical measurements were performed through capacitance-voltage using a Keithley 4200 impedance analyzer. 100 nm thick Ag film was evaporated as the back electrode, circular Ag electrode with 200 um diameter and 100 nm thickness was evaporated on target film through a shadow mask served as the front side electrode. Capacitance measurements were conducted at 500 kHz 100 mV ac modulation while the DC gate voltage was swept from −4 V to 4 V. DFT calculations were carried out through first-principles plane-wave pseudopotential formulation implemented in the Vienna ab-initio Simulation Package[59]. The exchange−correlation functional was in the form of Perdew−Burke−Ernzerhof with the generalized gradient approximation[60]. Van der Waals interactions were also considered using the DFT-D3 method[61]. A 6.5-Å-thick $SiO_2$ (001) surface with hydroxylation was built to resemble the experimental surface. The nudged elastic band method was used to locate the transition state between two local minima states. Gibbs correction was proposed for precursor adsorption using VASPKIT[62] at the temperature of 473 K.

### Reporting summary

Further information on research design is available in the Nature Portfolio Reporting Summary linked to this article.

## Data availability

All data in this study are available in the manuscript and in the Supplementary information section. Source data are provided with this paper.

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

## Acknowledgements

This work is supported by the National Key Research and Development Program of China (2022YFF1500400), the National Natural Science Foundation of China (51835005, 52273237) and the New Cornerstone Science Foundation through the XPLORER PRIZE. We would also like to acknowledge the technical support from the Analytic Testing Center and the Flexible Electronics Research Center of HUST.

## Author contributions

Y.L., K.C., and R.C. conceived the concept. Y.L. and Z.Q. performed experiments and the majority characterizations. J.Z., E.G., and J.Y. performed TEM and XPS characterizations. Y.L., Y.W., and B.S. performed DFT calculations. J.L. prepared the $Cu/SiO_2$ patterned samples from the chip production line and analyzed the TEM results. Y.L., K.C., and R.C. contributed to the paper writing.

## Competing interests

The authors declare no competing interests.
