## [Peer Review File · Nature Communications]

Reviewer comments, first round

Reviewer #1 (Remarks to the Author):

The authors interestingly describe achieving the selectivity Cu/SiO₂ patterned substrates using a co-reactant. Since the selectivity of Cu vs. SiO₂ increased by applying these process, it is envisaged to be applied to various fields using TaOx. However, there are a few parts that need more explanation.

1. The introduction has mainly covered references on the selective deposition. However, the reviewer thought that the authors should provide the detailed motivation of using TaOx films. In addition, have the authors tried another precursor for redox-coupled ALD of TaOx?
2. In comparison with figure 3a and figure 3c, the behavior of mass change upon H₂O exposure seems to be different. The authors should provide supporting information on this behavior. Is it just a difference in the scale range of the y-axis?
3. Both H₂O and EtOH are used as co-reactants for TaOx ALD. In redox-coupled TaOx ALD, the reviewer thought that EtOH could be attributed to the growth of TaOx on SiO₂. Are there any residual carbon issue from EtOH moieties in TaOx film?
4. The author should describe whether there is any change in the electrical properties of the TaOx thin films in applying the reduction-adsorption-oxidation process to TaOx ALD.
5. The authors should correct the typo (the chinese character -> cycles) in figure S4a.
6. In figure S4a and S4b, the TaOx film thickness at the particular ALD cycles seems to be lower than zero. I thought of the fitting error in SE analysis. The authors need to calibrate and fix SE datas to avoid critical comments.

Reviewer #2 (Remarks to the Author):

The research in the field of area-selective processes is of great interest for the fabrication of miniaturized chip components and on-chip interconnects as highlighted in the study. The self-aligned via patterning facilitated by area-selective deposition of a dielectric exclusively on dielectric surfaces in the presence of exposed copper metal is one of the examples where ASD processes may contribute to the chip miniaturization trend.

In the latter context, the authors demonstrate the possibility of inherently selective deposition of ALD TaOx on top of dielectric surfaces. The nucleation of TaOx on copper is modulated by the oxidation state of the Cu surface. The study focuses on the effect of oxidizing and reducing precursor pulses on the Cu surface and hence on the ALD TaOx process selectivity. Regular metal surface reduction using ethanol pulses is shown to enhance the selectivity of the standard two-step ALD TaOx process via suppressed adsorption of the Ta precursor on the oxide-free Cu.

As similar Cu surface oxidation/reduction steps considered in this work have been studied elsewhere, [1-2] the major novelty of the study is the actual ALD TaOx process (Ta(NtBU)(NEt₂)₃ / H₂O) and its three-step version which embeds a Cu surface reduction step via EtOH pulses. Although the study provides extensive experimental and theoretical evidence for most of the claims in the manuscripts there are a few major points of concern that need to be addressed.

1. According to TaOx thickness values shown in Figure 1d (section III), the proposed ABC-cycle (EtOH + Ta* + H₂O) shows a relatively small improvement in selectivity compared to the standard ALD process (Ta* + H₂O), which raises a question regarding the necessity of this process complication.

2. While the key application targeted by the proposed process is self-aligned via fabrication, TaOx explored in this work is certainly not an optimal material for integration into on-chip interconnects due to its very high dielectric constant ($k > 17-20$). Therefore, it would be good to demonstrate the possibility to extend the proposed ABC method to other more application-relevant dielectrics with lower k -value. For example, it could be done via simulation of different metal precursors and their adsorption on the oxide-free Cu surface as it is already done for the Ta precursor.

3. The Authors claim 100% selectivity of ~ 5 nm TaOx deposited on SiO₂. Yet, this statement is not very accurate since the selectivity depends on the sensitivity of the method used to measure the amount of material deposited on growth and non-growth surfaces. For example, some deposition of TaOx on the Cu surface during the three-step ALD process could be detected by QCM as shown in Figure S14b of the Supplementary Materials. In this regard, it would be helpful if the authors could add AFM topography images collected on the Cu surface after the TaOx deposition (currently the manuscript contains such images only before TaOx deposition or only for TaOx on SiO₂) since AFM tends to provide more details about the nucleation defects which cannot be sensed by ellipsometry.

4. The applicability/accuracy of the proposed nucleation model for fitting the experimental data (TaOx thickness measured by ellipsometry) should be clarified in the manuscript, in particular, regarding the TaOx growth on growth-inhibiting Cu surface where the initial film is discontinuous.

Few minor remarks:

- There is no detailed information in the experimental section about the pulse scheme of the ALD processes under study (for both AB and ABC processes).
- The section about the thin film characterization (in "Materials and Methods") should be extended with more details about the ellipsometry and AFM analysis.
- Figures in the Supplementary Materials have very low resolution, and some labels/legends are not readable.
- Figure S11: To make a fair comparison between different ASD studies targeting FSAV application, the legend should be extended with information on the type of dielectric and measurement technique used to assess the deposition selectivity. Additionally, please, check the color of data points in the plot and the corresponding legend – there appears to be some mismatch.

References

- [1] Peña, L. F., Veyan, J.-F., Todd, M. A., Derecskei-Kovacs, A., & Chabal, Y. J. (2018). Vapor-Phase Cleaning and Corrosion Inhibition of Copper Films by Ethanol and Heterocyclic Amines. *ACS Applied Materials & Interfaces*, 10(44), 38610–38620. <https://doi.org/10.1021/acsami.8b13438>
- [2] Lecordier, L., Herregods, S., & Armini, S. (2018). Vapor-deposited octadecanethiol masking layer on copper to enable area selective Hf₃N₄ atomic layer deposition on dielectrics studied by in situ spectroscopic ellipsometry. *Journal of Vacuum Science & Technology A*, 36(3), 031605. <https://doi.org/10.1116/1.5025688>

Reviewer #3 (Remarks to the Author):

Review of the manuscript "Self-Aligned Patterning of Tantalum Oxide on SiO₂/Cu through Redox-coupled Inherently Selective Atomic Layer Deposition" [Manuscript id NCOMMS-23-03931]

This manuscript reports the origin of the selectivity between Cu/SiO₂ during TaOx growth and a method to maximize the selectivity. The author revealed that the oxidation of the Cu substrate was the cause of the selectivity loss, and was able to increase the selectivity by adding a step of injecting a reducing agent to control it.

Although most details are acceptable and the authors have contributed to the improvements of the areal selective deposition process field, further consideration is required on the interpretation of the controversial experimental results and the clarity of the overall composition of the paper.

Therefore, I would like to suggest a major revision before acceptance. Specific comments are;

1. In Figs. 1 (b), S4 (a-b), S6(d), S8 (a-b), and S9 (a-b), The thickness of TaOx grown on the Cu substrate is negative. Is it a meaningful value?

2. In Figure S4 (a), there is a typo in the x-axis title.
3. Line 136, The author proposed a model in which new nuclei are generated through the dissociation of the nucleus, which is considered to be a significant factor because it is up to 400 times faster than the nucleation of non-defect sites as shown in Table 1. However, it is known that the integration of nuclei through the Ostwald ripening tends to occur spontaneously, so the author needs to explain how nuclei are dissociated.
4. Line 150, It seems to help the reader's understanding if a more detailed derivation process for eq (3) is described in supplementary materials and the n_i is described.
5. In Figure S6, The authors said that the value of the fitting parameter G_v was approximated to be equal to experimental GPC in the linear growth stage. However, although the GPCs of SiO_2 , Al_2O_3 , and HfO_2 look similar on the graph, the G_v value of Al_2O_3 is fitted to only about 60% of that of SiO_2 . So, it would be helpful to understand if the comparison between GPCs is provided.
6. The mass gain aspect is different when H_2O is injected in the AB-type process and the ABC-type process. What makes this difference?
7. Line 324, The authors said that there was no mushroom growth during TaO_x deposition. However, in the TEM cross-section image, it can be seen that the top of Cu is higher even after TaO_x deposition, and mushroom growth will not originally occur in this structure, so it should not be concluded that mushroom growth does not occur. In addition, since the β was obtained as 1 in Table 1, lateral growth should be active, so it looks to the two results are conflict.
8. The resolution of the figures included in the supplementary material is too low.

Reviewer #1 (Remarks to the Author):

The authors interestingly describe achieving the selectivity Cu/SiO₂ patterned substrates using a co-reactant. Since the selectivity of Cu vs. SiO₂ increased by applying these process, it is envisaged to be applied to various fields using TaO_x. However, there are a few parts that need more explanation.

1. The introduction has mainly covered references on the selective deposition. However, the reviewer thought that the authors should provide the detailed motivation for using TaO_x films. In addition, have the authors tried another precursor for redox-coupled ALD of TaO_x?

Author reply:

Thank you very much for the valuable suggestions. We have added the motivation of applying TaO_x films in the introduction. Other Ta precursors are also tried in this work, including Ta(NMe₂)₅ and Ta(OEt)₅. However, both the precursors need highly-active O₃ as co-reactant, thus the Cu substrate will be strongly oxidized which is harmful to the selective ALD process. Thus, Ta(N^tBu)(NEt₂)₃ have been adopted as an optimized precursor in this work. Moreover, five more different oxides deposition are performed here to demonstrate the generality of the redox-coupled ALD proposed in this work, including two Si precursors ((SiH₃[N(C₃H₇)₂], DIPAS) (SiH[N(CH₃)₂]₃, TDMAS)) and Nb(N^tBu)(NEt₂)₃, W(N^tBu)₂(NMe₂)₂, and Mo(NMe₂)₂(N^tBu)₂ precursors. The redox coupled ABC-type ALD is also favorable to obtaining high selectivity with these precursors, which shows the wide applicability of the redox-coupled ALD method.

The experimental details of Ta(NMe₂)₅ and Ta(OEt)₅ are shown below. For ALD process performed with Ta(NMe₂)₅, the co-reactants including H₂O and O₂ are not active enough to initiate the deposition of TaO_x. Thus, O₃ is chosen as a co-reactant. **Figure R1(a)**. The deposition of TaO_x on Cu quickly occurs since the oxidation of Cu by O₃. (**Figure R1(b)**). By regulating the pulse time of the precursor (2 seconds shortened to 1 second), there are only 10 cycles of nucleation delay on Cu, and the selectivity window is very narrow, as shown in **Figure R1(c)**. In addition, by shortening the co-reactant O₃ dosing time (2 seconds to 0.5 seconds), adjusting deposition temperature, there is also no nucleation delay on the Cu surface, as shown in **Figure R1 (d) (e)**. Then, the redox-coupled ALD is adopted for Ta(NMe₂)₅ precursors by adding an EtOH pulse before each AB-type ALD (**Figure R1 (f)**). The growth rate on Cu and on SiO₂ is similar, the selectivity is still not able to be improved. Thus, it is deduced that the process including O₃ is unfavorable for selectivity. **Figure R1 b-e**. As for Ta(OEt)₅, the co-reactants, including H₂O and O₂, are also not active enough for the deposition of TaO_x. Thus, O₃ is chosen as a co-reactant. If only using O₃ as co-reactant in AB-type ALD, it is found that there is a long nucleation delay on CuO surface in our previous work (Chemistry of Materials 2022 34 (20), 9013-9022) shown in **Figure R2**.

The results of AB and ABC type ALD of dielectrics using Si precursors ((SiH₃[N(C₃H₇)₂], DIPAS) (SiH[N(CH₃)₂]₃, TDMAS)) and Nb(N^tBu)(NEt₂)₃, W(N^tBu)₂(NMe₂)₂, and Mo(NMe₂)₂(N^tBu)₂ precursors are shown in **figure R10 and R11**.

Figure R1. (a) The ALD of $\text{Ta}(\text{NMe}_2)_5$ with different co-reactants, including H_2O , O_2 , and O_3 . The relationship between the deposition thickness of TaO_x thin films and the number of ALD cycles. The deposition temperature of $200\text{ }^\circ\text{C}$ and the pulses and purging processes are (b) 2-10-2-10s, (c) 1-10-2-10s, and (d) 2-10-0.5-10s, respectively; (e) The corresponding relationship between TaO_x film thickness and ALD cycle number at deposition temperature of $300\text{ }^\circ\text{C}$ with 2-10-2-10s. (f) The TaO_x film thickness as a function of ALD cycles by the pulse sequence of EtOH (2s)- $\text{Ta}(\text{NMe}_2)_5$ (2s)- O_3 (2s).

Figure R2. The ALD of $\text{Ta}(\text{OEt})_5$ with different co-reactants, including H_2O , O_2 , and O_3 . The TaO_x film thickness as a function of ALD cycles on SiO_2 , 10 nm ZnO , and 10 nm CuO substrates.

Modifications:

We have added related discussion in revised manuscript:

“Other Ta precursors with different coordinating groups were also exploited, including Ta(NMe₂)₅ and Ta(OEt)₅. However, both the precursors needed highly-active O₃ as co-reactant, thus the Cu substrate was strongly oxidized which was harmful to the selective ALD process. (Figure S28)”

The detailed discussions and data of ALD process of Ta(NMe₂)₅ and Ta(OEt)₅ is added to Supplementary Materials:

“Figure S28. (a) The ALD of Ta(NMe₂)₅ with different co-reactants, including H₂O, O₂, and O₃. H₂O and O₂ are not active enough to initiate growth of TaO_x at 200 °C. The ALD growth rate of Ta(NMe₂)₅ and O₃ with ALD processes (b) 2-10-2-10s, (c) 1-10-2-10s, and (d) 2-10-0.5-10s, respectively. By regulating the pulse time of the precursor (2 seconds shortened to 1 second), there are only 10 cycles of nucleation delay on Cu, and the selectivity window is narrow. (e) The corresponding relationship between TaO_x film thickness and ALD cycle number at deposition temperature of 300 °C with 2-10-2-10s. (f) The TaO_x film thickness as a function of ALD cycles by ABC-type ALD of EtOH (2s)- Ta(NMe₂)₅ (2s)-O₃ (2s). The growth rate on Cu and on SiO₂ is similar with no nucleation delay. For Ta(OEt)₅, the co-reactants, including H₂O and O₂, are also not active enough for the deposition of TaO_x. Using O₃ as co-reactant in AB-type ALD, it is found that there is a long nucleation delay on CuO surface in our previous work (Chemistry of Materials 2022 34 (20), 9013-9022).”

The motivation for TaO_x is added in introduction part:

“Tantalum oxide exhibits selective deposition on various oxides, with no observable growth on copper. TaO_x films are widely used as insulating layer for nanoelectronics, functional layer for memory devices, etc ⁴⁹⁻⁵²”

2. In comparison with figure 3a and figure 3c, the behavior of mass change upon H₂O exposure seems to be different. The authors should provide supporting information on this behavior. Is it just a difference in the scale range of the y-axis?

Author reply:

Thanks for the suggestions. The reason for the insignificant mass gain after water pulses in ABC type ALD process is due to the small amount of water can be adsorbed at the reaction sites on the surface. On the other hand, the introduction of an ethanol pretreatment step consumes a portion of the hydroxyl groups on the substrate surface and converts them into -OET (Chemistry of Materials 2013, 25, 4849), thereby reducing the amount of precursor adsorption in the ABC cycle compared to the AB cycle. (Figure R3) The adsorption of water is also influenced with deposition temperature, more H₂O molecules are physically adsorbed at lower temperature, the initial mass gain is larger after dosing H₂O at 100 °C (Figure R4).

Figure R3. The mass gain as a function of deposition time on SiO₂ substrate during AB-type (Ta(NⁱBu)(NEt₂)₃-H₂O) and ABC-type (EtOH-Ta(NⁱBu)(NEt₂)₃-H₂O) ALD cycles

Figure R4. The mass gain of TaO_x on Cu and SiO₂ during AB-type (Ta(NⁱBu)(NEt₂)₃-H₂O) ALD at (a) 100 °C and (a) 300 °C. The insets show the amplified mass gain at 100 and 300 °C.

Modifications:

The QCM measurements and related discussions are added into revised Supplementary Materials.

“The adsorption of water was also influenced with deposition temperature, more H₂O molecules were physically adsorbed at lower temperature, thus the initial mass gain was larger after dosing H₂O at 100 °C.”

3. Both H₂O and EtOH are used as co-reactants for TaO_x ALD. In redox-coupled TaO_x ALD, the reviewer thought that EtOH could be attributed to the growth of TaO_x on SiO₂. Are there any residual carbon issues from EtOH moieties in TaO_x film?

Author reply:

Thank you for the valuable comments. We have added XPS sputtering depth measurements of TaO_x film fabricated by Ta(NⁱBu)(NEt₂)₃-H₂O and EtOH-Ta(NⁱBu)(NEt₂)₃-H₂O ALD processes. After 300 s etching, the amount of surface C purities decreases, then reaches to a constant value. Both films

contain a small amount of C composition with similar concentration. The TaO_x film with AB-type ALD contains 9.1 at. % C composition, and the ABC-type film contains 8.1 at.% C. (Figure R5) We think that the residual C maybe caused by the co-reactant of H₂O, which is not very active to remove all the ligands of Ta precursor. The introduction of EtOH has minimal influence to the C composition.

Figure R5. The XPS sputter depth profile of (a) 30 nm-thick TaO_x film by AB-type (Ta(N^tBu)(NEt₂)₃-H₂O) ALD and (b) 20 nm-thick TaO_x film by ABC-type (EtOH-Ta(N^tBu)(NEt₂)₃-H₂O) ALD.

Modifications:

The XPS sputter depth measurements of 30 nm-thick TaO_x film fabricated by AB-type (Ta(N^tBu)(NEt₂)₃-H₂O) and 20 nm-thick TaO_x film fabricated by ABC-type (EtOH-Ta(N^tBu)(NEt₂)₃-H₂O) ALD processes are added in revised Supplementary Materials.

Related discussions are added into manuscript.:

“To compare the composition and dielectric constant of TaO_x films fabricated by AB-type (Ta(N^tBu)(NEt₂)₃-H₂O) and ABC-type (EtOH-Ta(N^tBu)(NEt₂)₃-H₂O) ALD, XPS sputter depth profiles were analyzed. The results showed that the ethanol used in the ABC-type ALD process did not influence the carbon concentration in the deposited film. (Figure S21)”

4. The author should describe whether there is any change in the electrical properties of the TaO_x thin films in applying the reduction-adsorption-oxidation process to TaO_x ALD.

Author reply:

Thanks for the valuable suggestions. The I-V and C-V tests are adopted on both AB-type and ABC-type ALD of TaO_x films. The electrical measurements were performed through capacitance-voltage using an Keithley 4200 impedance analyzer. First, 100 nm thick Ag was evaporated both as the back electrode and the top electrode. The measurements were conducted with a small AC signal of 100 mV and by sweeping the DC gate voltage from -4 to 4V. According to the series capacitance characteristics, there is the following formula: $\frac{1}{C_{max}} = \frac{1}{C_{TaOx}} + \frac{1}{C_{SiO2}}$, The capacitance of the TaO_x thin film layer is

determined by the thickness T of the oxide layer and the top electrode area S , that is, $EOT = \frac{\epsilon_{SiO_2} d_{ox}}{\epsilon_{ox}}$, $C_{TaOx} = \frac{S \epsilon_0 \epsilon_{TaOx}}{T}$. The k -value was calculated from capacitance C measured using the formula for parallel plate capacitor: $k = C \times d / (\epsilon_0 \times S)$, where ϵ_0 denotes vacuum permittivity, S denotes area of top 100 nm thick Ag electrodes and d denotes the thickness of the film. Through the C-V test (Figure R6), the K value of the tantalum oxide film of AB and ABC ALD process is calculated about 21.6 and 20.6, which is in accordance with the theoretical K value of tantalum oxide (20-25). The K value of AB and ABC type fabricated films is similar.

Figure R6. The C-V characteristic curves of tantalum oxide thin film in AB, ABC process, respectively.

Modifications:

The C-V test results are added in Supplementary Materials and related discussion is added in the revised manuscript:

“The dielectric constants of the tantalum oxide film for AB and ABC ALD processes were 21.6 and 20.6, respectively. The k values were obtained through CV test and the values were similar. (Figure S22)”

The detail of C-V test is added in the experimental section:

“The electrical measurements were performed through capacitance-voltage using an Keithley 4200 impedance analyzer. First, 100 nm thick Ag was evaporated both as the back electrode and the top electrode. The measurements were conducted with a small AC signal of 100 mV and by sweeping the DC gate voltage from -4 to $4V$.”

5. The authors should correct the typo (the chinese character -> cycles) in figure S4a.

Author reply:

Thanks for the remind. We have modified the mistake. (Figure R7)

Figure R7. The film thickness as a function of the ALD cycles on four substrates at 200 °C EtOH as co-reactant

6. In figure S4a and S4b, the TaOx film thickness at the particular ALD cycles seems to be lower than zero. I thought of the fitting error in SE analysis. The authors need to calibrate and fix SE datas to avoid critical comments.

Author reply:

Thanks for the comments. The film thickness lower than zero is reasonable since the EtOH can reduce the Cu native oxide layer, the phenomenon is also found in other references (Thin Solid Films 734, 138868 (2021); Vacuum 195, 110686 (2022); Journal of Materials Science: Materials in Electronics 32, 5442-5456 (2021); ACS Applied Materials & Interfaces, 10(44), 38610–38620). We also added experiments and SE measurements of the bare Cu substrates after EtOH, H₂O, O₃ pulses. The results show that the decrease of Cu surface thickness by ethanol pretreatment is mainly due to the reduction of a small amount of native oxide layer on the Cu surface. H₂O has minimal influence to the surface oxide layer, while O₃ could strongly oxidized the Cu surface that increase the surface native oxide layer. (Figure R19) In order to avoid misleading, we modified the y-axis to “Oxide film thickness”. (Figure R8 (Figure S3 in revised Supplementary Materials))

Figure R8. The film thickness as a function of the ALD cycles on four substrates at 200 °C by using three ABC-type ALD strategies, including (a) EtOH-Ta(N^tBu)(NEt₂)₃-H₂O, (b) CH₃COOH-Ta(N^tBu)(NEt₂)₃-H₂O and (c) H₂O-Ta(N^tBu)(NEt₂)₃-O₃.

Modifications:

The discussion of film thickness decreases to lower zero is added in revised manuscript, the experiments and SE measurements of bare Cu substrates after EtOH, H₂O, O₃ pulses are added in Supplementary Materials.

“It is found that the film thickness could be lower than zero when ethanol is utilized in the ALD process, which is reasonable since the EtOH can reduce the Cu native oxide layer. SE measurements of the bare Cu substrates after EtOH, H₂O, O₃ pulses are tested. The results show that the decrease of Cu surface thickness by ethanol pretreatment is about 0.5nm. H₂O has minimal influence to the surface oxide layer, while O₃ could strongly oxidized the Cu surface that increase the surface oxide layer.”

Reviewer #2 (Remarks to the Author):

The research in the field of area-selective processes is of great interest for the fabrication of miniaturized chip components and on-chip interconnects as highlighted in the study. The self-aligned via patterning facilitated by area-selective deposition of a dielectric exclusively on dielectric surfaces in the presence of exposed copper metal is one of the examples where ASD processes may contribute to the chip miniaturization trend.

In the latter context, the authors demonstrate the possibility of inherently selective deposition of ALD TaOx on top of dielectric surfaces. The nucleation of TaOx on copper is modulated by the oxidation state of the Cu surface. The study focuses on the effect of oxidizing and reducing precursor pulses on the Cu surface and hence on the ALD TaOx process selectivity. Regular metal surface reduction using ethanol pulses is shown to enhance the selectivity of the standard two-step ALD TaOx process via suppressed adsorption of the Ta precursor on the oxide-free Cu.

As similar Cu surface oxidation/reduction steps considered in this work have been studied elsewhere, [1-2] the major novelty of the study is the actual ALD TaOx process $(\text{Ta}(\text{NtBU})(\text{NEt}_2)_3 / \text{H}_2\text{O})$ and its three-step version which embeds a Cu surface reduction step via EtOH pulses. Although the study provides extensive experimental and theoretical evidence for most of the claims in the manuscripts there are a few major points of concern that need to be addressed.

1. According to TaOx thickness values shown in Figure 1d (section III), the proposed ABC-cycle $(\text{EtOH} + \text{Ta}^* + \text{H}_2\text{O})$ shows a relatively small improvement in selectivity compared to the standard ALD process $(\text{Ta}^* + \text{H}_2\text{O})$, which raises a question regarding the necessity of this process complication.

Author reply:

Thanks for the suggestions. In this work, it is found that surface state of Cu is critical for inherently selective ALD. Thus, we tried various co-reactants, deposition temperatures, and precursors. Moreover, the ABC-type ALD is adopted and optimized by changing the chemicals and the pulse orders. Cu surface oxidation/reduction steps have rarely been reported in the AS-ALD fields. Although H₂O used as co-reactant could achieve high selectivity, the ABC type ALD process is necessary, not only because of the higher selectivity obtained, but also the longer nucleation delay on the non-growth Cu region (Figure R9), which is very important and a more reliable process for FSAV.

Figure R9. The film thickness as a function of the ALD cycles on four substrates at 200 °C by using three ABC-type ALD strategies, including (a) EtOH-Ta(N^tBu)(NEt₂)₃-H₂O, (b) CH₃COOH-Ta(N^tBu)(NEt₂)₃-H₂O and (c) H₂O-Ta(N^tBu)(NEt₂)₃-O₃.

Modifications:

“In region III, the EtOH-Ta(NⁿBu)(NEt₂)₃-H₂O ABC-type process was the most preferred one. It achieved the highest selectivity and the longest nucleation delay on the non-growth Cu region, both were critical for a reliable self-aligned oxide stacking.”

2. While the key application targeted by the proposed process is self-aligned via fabrication, TaOx explored in this work is certainly not an optimal material for integration into on-chip interconnects due to its very high dielectric constant ($k > 17-20$). Therefore, it would be good to demonstrate the possibility to extend the proposed ABC method to other more application-relevant dielectrics with lower k-value. For example, it could be done via simulation of different metal precursors and their adsorption on the oxide-free Cu surface as it is already done for the Ta precursor.

Author reply:

Thanks for this valuable suggestion. We agree with the reviewer’s opinion that lower k-value materials are favored for FSAV application to reduce RC delay. SiO₂ is considered as a classical low-k material for FSAV. Unfortunately, the choice of Si precursors is very limited. We have tried two kinds of Si precursors ((SiH₃[N(C₃H₇)₂], DIPAS) (SiH[N(CH₃)₂]₃, TDMAS)). From the experiments, it is found that the redox-coupled ALD process could also increase the selectivity on some kinds of metal vs. SiO₂ surface. Si(OEt)₅ is adopted with ABC-cycle. The ABC type ALD can significantly extend the nucleation delay and achieve a high selectivity between SiO₂ and metal W. (**Figure R10**) On the other hand, it is also found that using TDMAS as a Si precursor, although SiO₂ is grown on both Cu and SiO₂ surfaces, but the growth rate on the Cu surface was largely reduced that increase the selectivity with redox coupled ALD using ethanol as reduction step. (**Figure R10**)

From the above experiments, we think that the method proposed in our work has potential to be applied to lower k value materials such SiO₂. The precursors, ALD parameters etc. could be tuned to improve the selectivity, and will be our future work.

Figure R10. The SiO_x film thickness as a function of ALD cycles by the DIPAS and TDMAS precursor.

Simulations are also performed for different Si precursors (TDMAS, BDEAS, TICS, TEOS, Si(NtBu)(NEt)₂, DIPAS). The absorption energy, reaction energy barrier for the most favorable path as well as the reaction rates on these two substrates are calculated (Table 1). It is found the differences of the reaction kinetics of different Si precursors on SiO₂/Cu is not as obvious as those of Ta precursors. Thus, we think developing new kinds of Si precursors is also very important in the field of selective ASD to extend the materials and applications.

In addition, other precursors with similar ligands compared with Ta(B^tBu)(NEt)₂ are adopted, including Nb(N^tBu)(NEt)₂, W(N^tBu)₂(NMe₂)₂, and Mo(NMe₂)₂(N^tBu)₂ to fabricate related dielectrics. The redox coupled ABC-type ALD is favorable to obtaining high selectivity compared with traditional AB-type ALD. The selectivity is confirmed by SE and XPS measurements (**Figure R11** (**Figure S24-26** in Supplementary Materials)). The reaction energy path of the above precursors on Cu and SiO₂ substrates are calculated via first-principles calculations. Table 2 shows the adsorption energies (ΔE for pre_ads), energy barriers along different steps (H transfer and HNEt₂ desorption). By comparing the energy barriers on SiO₂ and Cu, it can be found that the precursors of Mo, W and Nb exhibit much more favorable ALD reactions on SiO₂ than those on Cu substrate, which is similar with the Ta precursor. This trend implies that the selective ALD of NtBu related precursor on Cu/SiO₂ is robust for those metal oxides. **Table R2.** Although, the NbO_x, WO_x, MoO_x have high K value, the results could also demonstrate the wide adaptability of the redox coupled ALD method proposed in this work.

Table R1. Calculated energy for different reaction paths on Cu and SiO₂ substrate

Surface Precursor	SiO ₂			Cu		
	reaction	Eb	deltaE	reaction	Eb	deltaE
TDMAS	pre_ads	0.00	-0.69	pre_ads	0.00	-1.21
	ex_NMe2	0.88	-0.19	dec-H	0.01	-0.20
	NMe2_des		0.24	dec-NMe2	1.41	-0.31
Si_NtBu	pre_ads	0.00	-0.77	pre_ads	0.00	-2.85
	ex_HNEt2	0.13	-1.09	dec-NtBu	0.26	-0.77
	HNEt2_des	1.38	1.38	dec-NEt2	1.12	0.04
TICS	pre_ads	0.00	-0.35	pre_ads	0.00	-0.77
	ex_HCNO	1.91	-0.34	dec	0.67	-0.49
	HCNO_des	0.43	0.43			
BDEAS	pre_ads	0.00	-0.21	pre_ads	0.00	-1.34
	ex_HNEt2	0.15	-0.26	dec-NEt2	0.96	-0.25
	HNEt2_des	0.00	-0.02	dec-H	0.24	-0.74
TEOS	pre_ads	0.00	-0.53	pre_ads	0.00	-1.54
	ex_HOEt	0.95	-0.26	dec	1.23	0.47
	HOEt_des	0.43	0.43			
DIPAS	pre_ads	0.00	-0.24	pre_ads	0.00	-1.63
	ex_HNEt2	0.00	-0.26	dec-Nipr2	1.36	0.84
	HNEt2_des	0.00	-0.02	dec-H	0.05	-0.37

Table R2. Calculated energy for different reaction paths on Cu and SiO₂ substrate

Surface Precursor	SiO ₂			Cu		
	reaction	Eb	deltaE	reaction	Eb	deltaE
Ta (N ^t Bu)(NEt ₂) ₃	pre_ads	0.00	-0.75	pre_ads	0.00	-1.97
	H_trans1	0.37	-0.66	dec1	1.29	-0.76
	HNEt2_des1	0.22	-0.60	dec2	2.26	1.92
Mo(NMe ₂) ₂ (N ^t Bu) ₂	pre_ads	0.00	-0.50	pre_ads	0.00	-2.36
	H_trans1	0.33	-0.79	dec1	0.70	0.12
	HNEt2_des2	0.46	0.46	dec2	1.43	0.96
	H_trans1	0.46	-0.46			
	HNEt2_des2	0.80	0.80			
W(N ^t Bu) ₂ (NMe ₂) ₂	pre_ads	0.00	-0.39	pre_ads	0.00	-2.15
	H_trans1	0.37	-0.69	dec1	0.79	0.27
	HNEt2_des2	0.00	0.41	dec2	1.56	1.06
	H_trans1	0.59	-0.56			
	HNEt2_des2	0.74	0.74			
Nb(N ^t Bu)(NEt ₂) ₃	pre_ads	0.00	-0.97	pre_ads	0.00	-2.01
	H_trans1	0.03	-0.42	dec1	1.36	-0.71
	HNEt2_des1	0.00	-0.07	dec2	2.14	1.82

Figure R11. The film thickness as a function of the ALD cycles on SiO₂ and Cu substrates at 200°C by Nb(N^tBu)(NEt₂)₃, W(N^tBu)₂(NMe₂)₂, and Mo(NMe₂)₂(N^tBu)₂ precursors. The pulse sequences are (a) Nb(N^tBu)(NEt₂)₃ (2 s)-O₃ (2 s); (d) Nb(N^tBu)(NEt₂)₃ (2 s)-H₂O (0.5 s); (g) EtOH (2 s)-Nb(N^tBu)(NEt₂)₃ (2 s)-H₂O (0.5 s); (b) W(N^tBu)₂(NMe₂)₂ (2 s)-O₃ (2 s); (e) W(N^tBu)₂(NMe₂)₂ (2 s)-H₂O (0.5 s); (h) EtOH (2 s)- W(N^tBu)₂(NMe₂)₂ (2 s)-H₂O (0.5 s); (c) Mo(NMe₂)₂(N^tBu)₂ (2 s)-O₃ (2 s); (f) Mo(NMe₂)₂(N^tBu)₂ (2 s)-H₂O (0.5 s); (i) EtOH (2 s)- Mo(NMe₂)₂(N^tBu)₂ (2 s)-H₂O (0.5 s).

Modifications:

The discussions of other dielectrics fabricated with AB and ABC type ALD processes are added in revised manuscript.

“Several oxides, including NbO_x, MoO_x, and WO_x, have also been tried using AB and ABC-type ALD processes. All the results indicating that the redox-coupled ALD method is indeed capable to achieve higher selectivity compared to the binary ALD process, demonstrating the generality of this method. (Figure S24-26)”

The detail results of NbO_x, WO_x, MoO_x fabricated with AB and ABC type ALD processes are added in Supplementary Materials Figure S24-26.

3. The Authors claim 100% selectivity of ~5 nm TaO_x deposited on SiO₂. Yet, this statement is not very accurate since the selectivity depends on the sensitivity of the method used to measure the amount of material deposited on growth and non-growth surfaces. For example, some deposition of TaO_x on the Cu surface during the three-step ALD process could be detected by QCM as shown in Figure S14b of the Supplementary Materials. In this regard, it would be helpful if the authors could add AFM topography images collected on the Cu surface after the TaO_x deposition (currently the manuscript contains such images only before TaO_x deposition or only for TaO_x on SiO₂) since AFM tends to provide more details about the nucleation defects which cannot be sensed by ellipsometry.

Author reply:

Thanks for the valuable suggestion. To quantification of selectivity, XPS, AFM and cross-sectional TEM measurements are performed. As XPS is a highly surface sensitive characterization method, the amount of TaO_x deposited detected from XPS has high accuracy (Chem. Mater. 2018, 30, 663–670). High-resolution XPS confirms that no Ta 4f signal detected on Cu within 100 ALD cycles, indicating 100% selectivity. **(Figure R13)** Cross-sectional TEM images also demonstrate that no TaO_x on Cu regions, when the TaO_x film on SiO₂ is more than 5 nm. It is also indicated that deposition of TaO_x is only restricted on TaN barrier and SiO₂ regions. Diffusion of Ta atoms to the neighboring Cu regions is unlikely to be happened. **(Figure R14)** To detect nucleation defects, we also add AFM and SEM images on Cu after different ALD cycles as shown in **Figure R12**. Through SEM, the film is continuous and without large particles. Through AFM, the film is smooth and the roughness is maintained at a low value of ~0.7 nm, which is similar to the same batch of bare Cu deposited on Si wafer. Through the tests, there are no obvious nucleation defects on Cu regions before 100 ALD cycles.

Figure R12. SEM and AFM images of Cu substrates after 50, 100, and 150 ALD cycles. (a) SEM, (b) planar AFM and (c) 3-D AFM images after 50 ALD cycles; (d) SEM, (e) planar AFM and (f) 3-D AFM images after 100 ALD cycles; (g) SEM, (h) planar AFM and (i) 3-D AFM images after 150 ALD cycles

Figure R13. (a) The high-resolution XPS scan of Ta 4f and (b) the corresponding proportion of Ta element on Cu and SiO₂ after 50, 100, and 200 ALD cycles, respectively.

Figure R14. Cross-sectional TEM images of TaO_x deposited on SiO₂/Cu nanopatterned structures with ABC type ALD.

Although QCM shows some mass gain on Cu, the reason may be due to the different surface morphology of Cu on the QCM sensor and on Si wafer. (**Figure R15**) The surface of QCM sensor is highly rough compared with the smooth Cu films deposited on Si wafer. The roughness of Cu coated QCM sensor is ~30.6 nm, which is much higher than that on Cu deposited on Si wafer (~0.7 nm). The non-uniform surface structure may induce the adsorption of Ta precursor on surface defect and wrinkle sites which is hard to be purged, therefore causes a slow growth rate on Cu with QCM measurements. Although the QCM experimental results show that there is a small amount of precursor adsorption on the Cu surface under the same conditions, it is still a powerful tool to study the nucleation mechanisms of AB process and the ABC cycle.

Figure R15. (a-b) SEM and (c-d) AFM images for Cu film deposited on the QCM crystal oscillator

Modifications:

The measurements of XPS, AFM cross-sectional TEM and related discussions are added in revised manuscript and supplementary materials.

“High-precision X-ray photoelectron spectroscopy (XPS) was conducted to quantitatively compare the amount of TaO_x on Cu and SiO₂. At 50 and 100 ALD cycles, a peak ascribed to Ta 4f was not observed, indicating insignificant growth of TaO_x on Cu (inset in Figure 1c). The proportion of Ta was almost zero on the Cu surface at 50 cycles indicating 100% selectivity.”

“Through AFM tests, the Cu surfaces are smooth and the roughness is maintained at a low value of ~0.7 nm, which is similar to the bare Cu deposited on Si wafer.”

“There was no Ta element detected on copper region, while a strong Ta signal was detected on the SiO₂, which also confirmed the perfect selectivity of 100%. “

“High resolution TEM images of different regions also verified that TaO_x deposition was restricted to the TaN barrier and SiO₂, and diffusion of Ta atoms to neighboring Cu regions was unlikely to occur, avoiding excessive mushroom growth at the edges or the emergence of undesired nucleation defects within the Cu region. (Figure S24)”

The differences of QCM measurement and SE measurement on Si wafer are explained in revised manuscript.

“It should be noted that the growth rate on Cu surface is not suppressed to zero with QCM measurements, which may be caused by the Cu metrology deposited on crystal oscillator that has highly rough surface.”

4. The applicability/accuracy of the proposed nucleation model for fitting the experimental data (TaOx thickness measured by ellipsometry) should be clarified in the manuscript, in particular, regarding the TaOx growth on growth-inhibiting Cu surface where the initial film is discontinuous.

Author reply:

We are grateful for the reviewer's insightful suggestion to elaborate on the applicability and accuracy of our proposed nucleation model in fitting the experimental data. The model is an enhancement of the nucleation model proposed by Gregory N. Parsons (J. Vac. Sci. Technol. A, 2019, 37(2): 020911), which provides a solid foundation for describing the initial nucleation behavior of selective atomic layer deposition (ALD) on non-growth regions.

Parsons' model efficiently describes the selectivity and nucleation delay of selective ALD during discontinuous growth, incorporating aspects such as normal ALD nucleation, defect-induced nucleation, and expansion of existing nuclei. Our model takes this a step further by considering the net increase in nucleation sites due to atomic diffusion in and out of the dynamic expanding regions, specifically at the edges of the nuclei. This highlights the behavior in the initial nucleation stage induced by defects, ALD nucleation, and nucleus expansion, making it suitable to describe the growth of discontinuous films.

The film thickness in our model is derived from the average height of the volumes of all nuclei over the substrate area. This is then compared with experimental data, aligning well with the measured TaOx thickness by ellipsometry. It accurately fits the experimental data of selective growth on various substrates, including thickness, coverage, and nucleus numbers. For instance, **Figure R16** illustrates the fitting results of film thickness, coverage, and nucleus numbers for TiN selective ALD on an amorphous carbon substrate. These results have been previously reported in Chem. Mater. 30 (2018) 3223, our model aligns remarkably well with the experimental data, achieving a mean absolute error (MAE) of 0.081.

In the revised manuscript, we emphasize the model's applicability in describing the initial discontinuous nucleation on substrates. Additionally, to ensure the main manuscript remains concise, we have decided to relocate some of the details related to model construction to the Supplementary Information section. We believe this will aid the reader in focusing on the key findings and implications of our work.

Figure R16. The fitting results of TiN on carbon substrate using the nucleation model

Modifications:

We rewrote the paragraph on models in the article:

1) *“A nucleation model was developed to fit the selective ALD process, as indicated by the line curves in Figure 1b.”*

is revised as

“A nucleation model proposed by Parsons¹⁸ was adopted and varied to fit the selective ALD process, as indicated by the line curves in Figure 1b. The nucleation model includes the factors of the normal ALD nucleation (N^{\cdot} (nm^{-2})), defect induced nucleation (\hat{N} (nm^{-2})), the anisotropic growth of the existed nucleus, and atomic diffusion induced nucleation in the dynamic expanding region at the edge of nucleus (\hat{N}^{\prime} (nm^{-2})). Figure S9 The details of the nucleation model were described in Supplementary Materials. The fitting curves were consistent with the experimental results, and the proper error ferr was less than 2×10^{-2} . For SiO_2 and Al_2O_3 substrates, the values of nucleation site density induced per ALD cycle on non-defect sites N^{\cdot} ; defect-induced nucleation site density \hat{N} are at the order of 10^{-1} nm^{-2} .”

2) The details of the nucleation model are moved to SI

“The nucleation model includes five independent parameters: (1) the ALD growth rate of the material deposition on itself along the vertical direction per cycle \hat{G}_v (nm/cycle), (2) the ALD growth rate of the nuclei expansion on the substrate along the lateral direction per cycle \hat{G}_l (nm/cycle), (3) the nucleation site density induced per ALD cycle on non-defect sites \dot{N} (nm^{-2}), (4) the defect-induced nucleation site density during the initial ALD cycle \hat{N} (nm^{-2}), and (5) the atomic diffusion induced nucleation in the dynamic expanding region at the edge of nucleus \hat{N}^{\prime} (nm^{-2}). The perimeter of the growth zone represents the circumferential growth area around the nuclei. Growth along the lateral direction \hat{G}_l is characterized by the nuclei-substrate-precursor triple boundary and its rate is different from the intrinsic deposition rate. The parameter β is used to denote the ratio of \hat{G}_l to \hat{G}_v . The

remaining zones on the surface were free sites. The areas corresponding to these three zones after the j^{th} ALD cycle are denoted by A_{nuclei}^j , $A_{\text{perimeter}}^j$, and A_{free}^j , respectively. During the j^{th} cycle, the nuclei density can be described as follows:

$$N^j = \frac{A_{\text{free}}^{j-1}}{A} \left(\dot{N} + \frac{A_{\text{nuclei}}^{j-1}}{A} \dot{N}' \right) \quad (j \geq 2) \quad (1)$$

After the j^{th} cycle, the nucleation area includes two parts:

$$A_{\text{nuclei}}^{i \rightarrow j} = A_{\text{nuclei}}^{i \rightarrow (j-1)} + A_{\text{perimeter}}^{i \rightarrow (j-1)} \quad (1 \leq i \leq j-1) \quad (2)$$

Area of perimeter zone corresponds to nucleus expansion along the lateral direction, the evolution of which can be approximated by multiplying the lateral growth rate ($\dot{G}_l = \beta \dot{G}_v$) with the perimeter of the nuclei $L^{i \rightarrow (j-1)}$, and normalized by the fraction of free sites:

$$A_{\text{perimeter}}^{i \rightarrow j-1} = \beta \dot{G}_v \left(\frac{A_{\text{free}}^{j-2}}{A} \right) L^{i \rightarrow (j-1)} \quad (1 \leq i \leq j-1) \quad (3)$$

And the nuclei (seed generated during the i^{th} cycle) perimeter $L^{i \rightarrow j-1}$ after the $(j-1)^{\text{th}}$ cycle can be estimated from geometrical considerations. Here we estimate the nucleus is hemisphere, so the circular projection of the nucleus on substrate is circle, the relationship between the nucleus area and perimeter can be deduced:

$$A_{\text{nuclei}}^{i \rightarrow (j-1)} / n^i = \pi \left(\frac{L^{i \rightarrow (j-1)}}{2\pi n^i} \right)^2 \quad (4)$$

From which we re-write in the following form:

$$L^{i \rightarrow (j-1)} = 2\sqrt{\pi} \sqrt{A_{\text{nuclei}}^{i \rightarrow (j-1)} n^i} \quad (5)$$

The perimeter area for nuclei after the j^{th} cycle can be obtained as:

$$A_{\text{perimeter}}^{i \rightarrow j} = \beta \dot{G}_v \alpha \left(\frac{A_{\text{free}}^{j-1}}{A} \right) \sqrt{A_{\text{nuclei}}^{i \rightarrow j} n^i}, \quad \alpha = 2\sqrt{\pi} \quad (6)$$

The area of the free site zone after the j^{th} cycle decreases owing to the accumulation of j cycles of ALD growth:

$$A_{\text{free}}^j = A - \sum_{i=1}^j (A_{\text{nuclei}}^{i \rightarrow j} + A_{\text{perimeter}}^{i \rightarrow j}) \quad (7)$$

Through the recurrence relations, it is straightforward to obtain the value of $A_{\text{nuclei}}^{i \rightarrow j}$, $A_{\text{perimeter}}^{i \rightarrow j}$, and A_{free}^j iteratively with the initial value of \hat{N} , \dot{N} , \dot{N}' , \dot{G}_v , and β . Such model is tested to well fit the experimental data [Chem. Mater. 30 (2018) 3223] of the selectively growth on carbon substrates with MAE=0.081.”

3) “(5)the nucleation site density induced by the dissociation of the nucleus to the non-growth region \dot{N}' (nm^{-2}).”

is modified as

“(5) the atomic diffusion induced nucleation in the dynamic expanding region at the edge of nucleus N^{\wedge} (nm^{-2})”

Few minor remarks:

- There is no detailed information in the experimental section about the pulse scheme of the ALD processes under study (for both AB and ABC processes).

Author reply:

Thanks very much for the suggestions. We have added the details of the ALD processes. The heating temperature of the Ta precursor steel bubbler was 65 °C. The high-purity Ar gas (40sccm, 99.999%) was injected into the bubbler and carried precursors dosed into the ALD cavity. The pipeline temperature was up to 90 °C to prevent the precursors’ condensation.

Meanwhile, additional experiments are added to explore the influence of pulse time of Ta precursor to selectivity. The saturate time of Ta precursor on SiO₂ was ~2 s. Decreasing the Ta precursor pulse time to 1s, the nucleation delay on Cu could be maintained to 100 cycles while the growth rate on SiO₂ was too slow. Increasing the pulse time to 3s, the selectivity deteriorated due to quick nucleation on Cu. For ABC type ALD, The optimized EtOH pulse time is 2 s. The optimized pulse sequence is EtOH (2 s), Ar (10 s), Ta(NⁱBU)(NEt₂)₃ (2 s), Ar (10 s), H₂O (0.5 s), and Ar (10 s). **Figure R17 (Figure S7 in Supplementary Materials)**

Figure R17. The TaO_x film thickness as a function of the number of ALD cycles, and the pulse processes are (a) 1-10-0.5-10, (b) 2-10-0.5-10, and (c) 3-10-0.5-10, respectively, in seconds; (d) The corresponding relationship between TaO_x film thickness and Ta(NⁿBU)(NEt₂)₃ pulse time at the deposition temperature of 200 °C

Modifications:

“The heating temperature of the Ta precursor steel bubbler was 65 °C. The high-purity Ar gas (40sccm, 99.999%) was injected into the bubbler and carried precursors dosed into the ALD cavity. The pipeline temperature was up to 90 °C to prevent the precursors’ condensation.”

“The saturate time of Ta precursor on SiO₂ is ~2 s. Decrease the pulse time to 1s, the selectivity deteriorates due to low growth speed on SiO₂. While increasing the pulse time to 3s, the selectivity also deteriorates due to quick nucleation on Cu. The pulse and purge time of H₂O is saturated for self-limited reaction. The optimized pulse sequence is EtOH (2 s), Ar (10 s), Ta(NⁿBU)(NEt₂)₃ (2 s), Ar (10 s), H₂O (0.5 s), Ar (10 s).”

- The section about the thin film characterization (in “Materials and Methods”) should be extended with more details about the ellipsometry and AFM analysis.

Author reply:

Thanks for the suggestions. We have added details about the ellipsometry and AFM analysis. For example, the SE tests are fitted by a modified TaO_x Cauchy model. And AFM is performed in tapping mode using a Molecular Imaging PicoScan Controller.

Modifications:

“The modified TaO_x Cauchy model was used to fit the ellipsometer data through the software of Complete EASE. J.A.Woollam M2000 spectroscopic ellipsometry is utilized to collect data.”

“AFM (SPM-9700HT, Shimadzu) was performed in tapping mode using a Molecular Imaging PicoScan Controller. Aluminum reflex-coated Si AFM probes were used. Data were processed and analyzed using Gwyddion 2.49 software.”

- Figures in the Supplementary Materials have very low resolution, and some labels/legends are not readable.

Author reply:

Thanks for the suggestions. We have modified all the figures to improve the resolution to 600 dpi. Other modifications of the Supplementary Materials are listed below.

Modifications:

In order to avoid misleading, for example, considering that the oxide layer on the copper surface is reduced or etched, we have changed the y-axis coordinates of some figures (S3(a)-(b), S8, S10(d), and S12, S13) to “Oxide film thickness”.

We have added **Figure S6-9, S19, S21-28** for new experimental results, and added relevant references of **Figure S15**.

To detect nucleation defects, we added AFM and SEM images on Cu after different ALD cycles as shown in **Figure S6**.

Nucleation modeling part was added in **Figure S8**.

To verify different regions of the structure sample, the copper surface diffusion phenomenon was seen similarly was added in **Figure S23-24**.

Another precursor for redox-coupled ALD of TaOx is added, and the experimental details of Ta(NMe₂)₅ and Ta(OEt)₅ were shown in **Figure S28**.

To understand why the redox coupled ABC-type ALD is favorable to obtaining high selectivity compared with traditional AB-type ALD, various precursors with similar ligands compared with Ta(BⁱBu)(NEt₂)₅ were adopted in **Figure S25-27**, including Nb(NⁱBu)(NEt₂)₃, W(NⁱBu)₂(NMe₂)₂, and Mo(NMe₂)₂(NⁱBu)₂.

To figure out the reason for the insignificant mass gain after water pulses using ABC type ALD process, the mass gain as a function of deposition time and temperature on different substrate were added in **Figure S17-18**.

To figure out if there are any residual carbon issues from EtOH moieties in TaOx film, we added the XPS sputter depth measurements of TaOx film by Ta(NⁱBu)(NEt₂)₃-H₂O and EtOH-Ta(NⁱBu)(NEt₂)₃-H₂O ALD processes in **Figure S21**.

To figure out if there is any change in the electrical properties of the TaOx thin films in applying the reduction-adsorption-oxidation process to TaOx ALD, we added the C-V tests of both AB-type and ABC-type ALD of TaOx films in **Figure S22**.

The film thickness as a function of the ALD cycles on four substrates at 200 °C by using three ABC-type ALD strategies was added in **Figure S12**.

To figure out the surface morphology of QCM crystal oscillator were added in **Figure S19**.

In order to verify the influence of three co-reactants on the Cu, the surface of cu substrate was treated with ethanol, water and ozone at 200 °C, which was added in **Figure S8**.

- Figure S11: To make a fair comparison between different ASD studies targeting FSAV application, the legend should be extended with information on the type of dielectric and measurement technique used to assess the deposition selectivity. Additionally, please, check the color of data points in the plot and the corresponding legend – there appears to be some mismatch.

Author reply:

Thanks for the suggestions. We have added information on the type of dielectric and measurement technique. The color of the data points and the corresponding legend are also modified. **Figure R18 (Figure S15 in Supplementary Materials)**

Modifications:

Please see the modified “Figure S15”

Figure R18. The trade-off between the selectivity and film thickness on SiO₂. (ref. 1-11)

References

- [1] Peña, L. F., Veyan, J.-F., Todd, M. A., Derecskei-Kovacs, A., & Chabal, Y. J. (2018). Vapor-Phase Cleaning and Corrosion Inhibition of Copper Films by Ethanol and Heterocyclic Amines. *ACS Applied Materials & Interfaces*, 10(44), 38610–38620. <https://doi.org/10.1021/acsami.8b13438>
- [2] Lecordier, L., Herregods, S., & Armini, S. (2018). Vapor-deposited octadecanethiol masking layer on copper to enable area selective Hf 3 N 4 atomic layer deposition on dielectrics studied by in situ spectroscopic ellipsometry. *Journal of Vacuum Science & Technology A*, 36(3), 031605. <https://doi.org/10.1116/1.5025688>

Author reply:

Thanks for the suggestions. We have added the related references and strengthened the related discussions.

Modifications:

“23 Lecordier, L., Herregods, S., Armini, S. Vapor-deposited octadecanethiol masking layer on copper to enable area selective Hf 3 N 4 atomic layer deposition on dielectrics studied by in situ spectroscopic ellipsometry. *J. Vac. Sci. Technol. A*, 36(3): 031605 (2018).

45 Pena, L. F., Veyan, J. F., Todd, M. A., Derecskei-Kovacs, A. & Chabal, Y. J. Vapor-Phase Cleaning and Corrosion Inhibition of Copper Films by Ethanol and Heterocyclic Amines. *ACS Appl. Mater. Interfaces* 10, 38610-38620(2018).

”

Reviewer #3 (Remarks to the Author):

Review of the manuscript “Self-Aligned Patterning of Tantalum Oxide on SiO₂/Cu through Redox-coupled Inherently Selective Atomic Layer Deposition” [Manuscript id NCOMMS-23-03931]

This manuscript reports the origin of the selectivity between Cu/SiO₂ during TaOx growth and a method to maximize the selectivity. The author revealed that the oxidation of the Cu substrate was the cause of the selectivity loss, and was able to increase the selectivity by adding a step of injecting a reducing agent to control it.

Although most details are acceptable and the authors have contributed to the improvements of the areal selective deposition process field, further consideration is required on the interpretation of the controversial experimental results and the clarity of the overall composition of the paper. Therefore, I would like to suggest a major revision before acceptance. Specific comments are;

1. In Figs. 1 (b), S4 (a-b), S6(d), S8 (a-b), and S9 (a-b), The thickness of TaOx grown on the Cu substrate is negative. Is it a meaningful value?

Author reply:

Thanks for the comments. The film thickness lower than zero is reasonable since the EtOH can reduce the Cu native oxide layer, the phenomenon is also found in other references (Thin Solid Films 734, 138868 (2021); Vacuum 195, 110686 (2022); Journal of Materials Science: Materials in Electronics 32, 5442-5456 (2021); ACS Applied Materials & Interfaces, 10(44), 38610–38620). We also added experiments and SE measurements of the bare Cu substrates after EtOH, H₂O, O₃ pulses (2 s pulse with 10 s Ar purging). The results show that the decrease of Cu surface thickness by ethanol pretreatment is mainly due to the reduction of a small amount of native oxide layer on the Cu surface. About 0.5nm native oxide of Cu is reduced measured with SE. H₂O has minimal influence to the surface oxide layer, while O₃ could strongly oxidized the Cu surface that increase the surface native oxide layer about 1.5-2nm. **(Figure R19)** In order to avoid misleading, we modified the y-axis to “Oxide film thickness” in the figure with EtOH pretreatment.

Figure R19. The surface of cu substrate was treated with ethanol, water and ozone at 200 °C

Modifications:

“It is found that the film thickness could be lower than zero when ethanol is utilized in the ALD process, which is reasonable since the EtOH can reduce the Cu native oxide layer. SE measurements of the bare Cu substrates after EtOH, H₂O, O₃ pulses are tested. The results show that the decrease of Cu surface

thickness by ethanol pretreatment is about 0.5nm. H₂O has minimal influence to the surface oxide layer, while O₃ could strongly oxidized the Cu surface that increase the surface oxide layer.”

The data of cu substrate treated with ethanol, water and ozone at 200 °C are added in Supplementary Materials figure S8.

2. In Figure S4 (a), there is a typo in the x-axis title.

Author reply:

We have modified the typo. **Figure R7 (Figure S3 in Supplementary Materials)**

3. Line 136, The author proposed a model in which new nuclei are generated through the dissociation of the nucleus, which is considered to be a significant factor because it is up to 400 times faster than the nucleation of non-defect sites as shown in Table 1. However, it is known that the integration of nuclei through the Ostwald ripening tends to occur spontaneously, so the author needs to **explain how nuclei are dissociated.**

Author reply:

We appreciate the reviewer's insightful comments and the opportunity to clarify our model. We agree that our original term "dissociation" may have led to some confusion. In our model, the term was used to denote the net increase of nucleation sites due to atomic diffusion into and out of the dynamic expansion region (near the edge of a nucleus). We have revised our manuscript to ensure this definition is more clearly articulated.

This concept is consistent with multiple experimental studies that have observed changes in nucleation induced by diffusion during the dynamic growth of the nucleus. Some studies have reported an increase in nucleation (J. Electrochem. Soc., 1989, 136(1): 271; Thin Solid Films, 1994, 241(1-2): 310-317; J. Phys. Chem. C, 2017, 121(11): 5871-5881), while others have shown a decrease due to ripening (Chem. Mater., 2020, 32(22): 9560-9572; J. Phys. Chem. Lett., 2017, 8(5): 975-983).

In our model, the parameters \dot{N}' and \dot{N} represent nucleation probabilities. A higher value of \dot{N}' relative to \dot{N} implies that the probability of diffusion-induced nucleation is much higher than that of normal ALD nucleation on non-growth regions. This is indicative of the probability difference, rather than a difference in deposition rates. Our model suggests that decreasing the ratio \dot{N}'/\dot{N} (\dot{N} fixed) from 400 to 40 changes the nucleation delay from 52 cycles to 71 cycles, indicating that an increase in normal ALD nucleus generation accelerates nucleation on non-growth regions.

We apologize for any misunderstanding caused by our initial description. In the revised manuscript, we have clarified the meanings of \dot{N}' and \dot{N} included additional details of the model and supporting references in the Supplementary Information.

Modifications:

“(5) the nucleation sites density induced by the dissociation of the nucleus to the non-growth region” is modified as

“(5) the atomic diffusion induced nucleation in the dynamic expanding region at the edge of nucleus $N' \text{ (nm}^{-2}\text{)}$ ”

4. Line 150, It seems to help the reader's understanding if a more detailed derivation process for eq (3) is described in supplementary materials and the n_i is described.

Author reply:

We appreciate the reviewer's careful check about the details of the model. The n_i is the total number of seeds generated on the j^{th} cycle. Derivation of Eq (3) is presented as follows:

Area of perimeter zone corresponds to nucleus expansion along the lateral direction, the evolution of which can be approximated by multiplying the lateral growth rate ($\dot{G}_l = \beta\dot{G}_v$) with the perimeter of the nuclei $L^{i \rightarrow (j-1)}$, and normalized by the fraction of free sites:

$$A_{\text{perimeter}}^{i \rightarrow j-1} = \beta\dot{G}_v \left(\frac{A_{\text{free}}^{j-2}}{A} \right) L^{i \rightarrow (j-1)} \quad (1 \leq i \leq j-1) \quad (3)$$

And the nuclei (seed generated during the i^{th} cycle) perimeter $L^{i \rightarrow j-1}$ after the $(j-1)^{\text{th}}$ cycle can be estimated from geometrical considerations. Specifically, we assume the nuclei adopt a hemispherical shape. Consequently, the nuclei's projection onto the substrate forms a circular shape. This assumption allows us to deduce the relationship between the nucleus area and its perimeter:

$$A_{\text{nuclei}}^{i \rightarrow (j-1)} / n^i = \pi \left(\frac{L^{i \rightarrow (j-1)}}{2\pi n^i} \right)^2 \quad (4)$$

From which we re-write in the following form:

$$L^{i \rightarrow (j-1)} = 2\sqrt{\pi} \sqrt{A_{\text{nuclei}}^{i \rightarrow (j-1)} n^i} \quad (5)$$

Substituting the expression of $L^{i \rightarrow (j-1)}$ into Eqn **Error! Reference source not found.**, the relationship between nuclei perimeter zone and nucleation zone after $(j-1)^{\text{th}}$ cycle for nuclei generated during the i^{th} ALD cycle is:

$$A_{\text{perimeter}}^{i \rightarrow (j-1)} = \beta\dot{G}_v \alpha \left(\frac{A_{\text{free}}^{j-2}}{A} \right) \sqrt{A_{\text{nuclei}}^{i \rightarrow (j-1)} n^i}, \quad \alpha = 2\sqrt{\pi} \quad (6)$$

In revised manuscript, we supplemented the description of n_i and derivation process in the SI according to the reviewer's suggestion.

Modifications:

Nucleation Modeling part is added in SI

“The nucleation model includes five independent parameters: (1) the ALD growth rate of the material deposition on itself along the vertical direction per cycle \dot{G}_v (nm/cycle), (2) the ALD growth rate of the nuclei expansion on the substrate along the lateral direction per cycle \dot{G}_l (nm/cycle), (3) the nucleation site density induced per ALD cycle on non-defect sites \dot{N} (nm^{-2}), (4) the defect-induced nucleation site density during the initial ALD cycle \hat{N} (nm^{-2}), and (5) the atomic diffusion induced nucleation in the dynamic expanding region at the edge of nucleus \dot{N}' (nm^{-2}). The perimeter of the growth zone represents the circumferential growth area around the nuclei. Growth along the lateral

direction \dot{G}_l is characterized by the nuclei-substrate-precursor triple boundary and its rate is different from the intrinsic deposition rate. The parameter β is used to denote the ratio of \dot{G}_l to \dot{G}_v . The remaining zones on the surface were free sites. The areas corresponding to these three zones after the j^{th} ALD cycle are denoted by A_{nuclei}^j , $A_{\text{perimeter}}^j$, and A_{free}^j , respectively. During the j^{th} cycle, the nuclei density can be described as follows:

$$N^j = \frac{A_{\text{free}}^{j-1}}{A} \left(\dot{N} + \frac{A_{\text{nuclei}}^{j-1}}{A} \dot{N}' \right) \quad (j \geq 2) \quad (1)$$

After the j^{th} cycle, the nucleation area includes two parts:

$$A_{\text{nuclei}}^{i \rightarrow j} = A_{\text{nuclei}}^{i \rightarrow (j-1)} + A_{\text{perimeter}}^{i \rightarrow (j-1)} \quad (1 \leq i \leq j-1) \quad (2)$$

Area of perimeter zone corresponds to nucleus expansion along the lateral direction, the evolution of which can be approximated by multiplying the lateral growth rate ($\dot{G}_l = \beta \dot{G}_v$) with the perimeter of the nuclei $L^{i \rightarrow (j-1)}$, and normalized by the fraction of free sites:

$$A_{\text{perimeter}}^{i \rightarrow j-1} = \beta \dot{G}_v \left(\frac{A_{\text{free}}^{j-2}}{A} \right) L^{i \rightarrow (j-1)} \quad (1 \leq i \leq j-1) \quad (3)$$

And the nuclei (seed generated during the i^{th} cycle) perimeter $L^{i \rightarrow j-1}$ after the $(j-1)^{\text{th}}$ cycle can be estimated from geometrical considerations. Here we estimate the nucleus is hemisphere, so the circular projection of the nucleus on substrate is circle, the relationship between the nucleus area and perimeter can be deduced:

$$A_{\text{nuclei}}^{i \rightarrow (j-1)} / n^i = \pi \left(\frac{L^{i \rightarrow (j-1)}}{2\pi n^i} \right)^2 \quad (4)$$

From which we re-write in the following form:

$$L^{i \rightarrow (j-1)} = 2\sqrt{\pi} \sqrt{A_{\text{nuclei}}^{i \rightarrow (j-1)} n^i} \quad (5)$$

The perimeter area for nuclei after the j^{th} cycle can be obtained as:

$$A_{\text{perimeter}}^{i \rightarrow j} = \beta \dot{G}_v \alpha \left(\frac{A_{\text{free}}^{j-1}}{A} \right) \sqrt{A_{\text{nuclei}}^{i \rightarrow j} n^i}, \quad \alpha = 2\sqrt{\pi} \quad (6)$$

The area of the free site zone after the j^{th} cycle decreases owing to the accumulation of j cycles of ALD growth:

$$A_{\text{free}}^j = A - \sum_{i=1}^j (A_{\text{nuclei}}^{i \rightarrow j} + A_{\text{perimeter}}^{i \rightarrow j}) \quad (7)$$

Through the recurrence relations, it is straightforward to obtain the value of $A_{\text{nuclei}}^{i \rightarrow j}$, $A_{\text{perimeter}}^{i \rightarrow j}$, and A_{free}^j iteratively with the initial value of \hat{N} , \dot{N} , \dot{N}' , \dot{G}_v , and β . Such model is tested to well fit the experimental data [Chem. Mater. 30 (2018) 3223] of the selectively growth on carbon substrates with MAE=0.081."

5. In Figure S6, The authors said that the value of the fitting parameter G_v was approximated to be equal to experimental GPC in the linear growth stage. However, although the GPCs of SiO_2 , Al_2O_3 , and HfO_2 look similar on the graph, the G_v value of Al_2O_3 is fitted to only about 60% of that of SiO_2 . So, it would be helpful to understand if the comparison between GPCs is provided.

Author reply:

We appreciate the reviewer's insightful suggestion to compare growth per cycle (GPC) values. In Table S1 of the Supplementary Information, we compiled the thicknesses of TaO_x on different substrates along with the ratios of GPCs for all samples. The first row of the table presents the experimental values corresponding to the samples depicted in Figure S6.

Upon analysis, we found that the experimental GPC ratio of TaO_x on Al_2O_3 to SiO_2 is approximately 76%, which aligns closely with the 62% ratio derived from our model's fitting result. When considering other samples, the ratios of fitted G_v values also correspond well with the experimental values. This correlation supports the suitability of our proposed model in describing the nucleation behavior of selective atomic layer deposition (ALD). (**Table R1**)

In response to your suggestion, we have included a comparison of GPCs, both from experimental data and fitting results, in the revised manuscript. We believe this additional detail will help readers better understand the application and accuracy of our model.

Table R1 The thicknesses (nm) of TaO_x on different substrates and GPC ratios

Sample	SiO_2	Al_2O_3	HfO_2	GPC Ratio $\text{Al}_2\text{O}_3/\text{SiO}_2$	Fitting G_v ratio $\text{Al}_2\text{O}_3/\text{SiO}_2$
Ta-H₂O 200°C 100 cycles	11.22	8.525	8.115	76.0%	62.3%
Ta- 2 EtOH 00°C 100 cycles	1.52	0.855	0.645	56.3%	82.1%
Ta-HAc 200°C 100 cycles	2.955	2.285	2.195	77.3%	75.0%
Ta-O ₃ 200°C 100 cycles	8.505	6.755	6.1	79.4%	72.9%
EtOH-Ta-H ₂ O 200°C 100 cycles	3.64	3.005	2.575	82.6%	62.2%
HAc-Ta-H ₂ O 200°C 100 cycles	8.52	7.33	6.75	86.0%	89%
H ₂ O-Ta-O ₃ 200°C 100 cycles	11.8	9.755	8.235	82.7%	62.3%

Modifications:

“The G_v value of TaO_x on Al_2O_3 is fitted to be about 62% that of SiO_2 in Figure S6, which agrees with the experimental ratio 78% (Table S1)”

1) We add Table S1 in the SI

2) Table S1 The thicknesses (nm) of TaO_x on different substrates and GPC ratios

Sample	SiO_2	Al_2O_3	HfO_2	GPC Ratio $\text{Al}_2\text{O}_3/\text{SiO}_2$	Fitting G_v ratio $\text{Al}_2\text{O}_3/\text{SiO}_2$
Ta-H₂O 200°C 100 cycles	11.22	8.525	8.115	76.0%	62.3%
Ta-EtOH 200°C 100 cycles	1.52	0.855	0.645	56.3%	82.1%
Ta-HAc 200°C 100 cycles	2.955	2.285	2.195	77.3%	75.0%

Ta-O ₃ 200°C 100 cycles	8.505	6.755	6.1	79.4%	72.9%
EtOH-Ta-H ₂ O 200°C 100 cycles	3.64	3.005	2.575	82.6%	62.2%
HAc-Ta-H ₂ O 200°C 100 cycles	8.52	7.33	6.75	86.0%	89%
H ₂ O-Ta-O ₃ 200°C 100 cycles	11.8	9.755	8.235	82.7%	62.3%

6. The mass gain aspect is different when H₂O is injected in the AB-type process and the ABC-type process. What makes this difference?

Author reply:

Thanks for the suggestions. The reason for the insignificant mass gain after water pulses is due to the small amount of water can be adsorbed at the reaction sites on the surface when ABC type ALD process is performed. On the other hand, the introduction of an ethanol pretreatment step consumes a portion of the hydroxyl groups on the substrate surface and converts them into -OET (Chemistry of Materials 2013, 25, 4849), thereby reducing the amount of precursor adsorption in the ABC cycle compared to the AB cycle. (Figure R3) The adsorption of water is also influenced with deposition temperature, more H₂O molecules are physically adsorbed at lower temperature, the initial mass gain is larger after dosing H₂O at 100 °C (Figure R4).

Modifications:

The QCM measurements and related discussions are added into revised Supplementary Materials.

“By the way, there are no peaks during H₂O pulse, which is caused by the less adsorbed precursors compared with AB-type ALD. Small amount of water can be adsorbed at the reaction sites on the surface when ABC type ALD process. The adsorption of water is also influenced with deposition temperature, more H₂O molecules are physically adsorbed at lower temperature, the initial mass gain is larger after dosing H₂O at 100 °C.”

7. Line 324, The authors said that there was no mushroom growth during TaO_x deposition. However, in the TEM cross-section image, it can be seen that the top of Cu is higher even after TaO_x deposition, and mushroom growth will not originally occur in this structure, so it should not be concluded that mushroom growth does not occur. In addition, since the β was obtained as 1 in Table 1, lateral growth should be active, so it looks to the two results are conflict.

Author reply:

We appreciate the reviewer's insightful comments and questions regarding the potential for mushroom growth during TaO_x deposition. To address these concerns, we have included additional cross-sectional TEM measurements of the TaO_x deposition process in the revised manuscript (see Figure R12).

These TEM images cover various regions of the Cu/SiO₂ substrate and clearly demonstrate that TaO_x deposition is exclusively localized on the TaN barrier and SiO₂. There's no evidence of Ta atoms diffusing into neighboring Cu regions or of significant mushroom growth occurring.

We would also like to clarify that the mushroom growth phenomenon does not directly depend on

the anisotropic growth parameter, β . This parameter characterizes the microscopic shape of the nucleus on the non-growth substrate (Cu), whereas mushroom growth relates to the excessive deposition on the active substrate (SiO_2), stemming from differences in ALD deposition rates between the two substrates. Therefore, these two factors are not directly correlated, and the observation of no mushroom growth does not conflict with the value of β on the Cu substrate.

In the revised manuscript, we have included additional cross-sectional TEM evidence to support our assertion of no mushroom growth on SiO_2/Cu nanopatterned structures.

Figure R20. Cross-sectional TEM images of TaO_x deposited on SiO_2/Cu nanopatterned structures with ABC type ALD.

Modifications:

1) The TEM images cover various regions of the Cu/SiO_2 substrate are added in supplementary material, related discussion is added in revised manuscript.

“Different regions are tested with TEM, it is demonstrated that the deposition of TaO_x is only restricted on TaN barrier and SiO_2 . Diffusion of Ta atoms to the neighboring Cu regions and obvious mushroom growth are unlikely to be happened. (Figure S20,21).”

2) Table 1 has been moved in SI as Table S2

8. The resolution of the figures included in the supplementary material is too low.

Author reply:

Thanks for the suggestions. We have modified all the figures to improve the resolution to 600 dpi. Other modifications of the Supplementary Materials are listed below.

Modifications:

In order to avoid misleading, for example, considering that the oxide layer on the copper surface is reduced or etched, we have changed the y-axis coordinates of some figures (**S3(a)-(b)**, **S8**, **S10(d)**, and **S12, S13**) to “Oxide film thickness”.

We have added **Figure S6-9, S19, S21-28** for new experimental results, and added relevant references of **Figure S15**.

*To detect nucleation defects, we added AFM and SEM images on Cu after different ALD cycles as shown in **Figure S6**.*

*Nucleation modeling part was added in **Figure S8**.*

*To verify different regions of the structure sample, the copper surface diffusion phenomenon was seen similarly was added in **Figure S23-24**.*

*Another precursor for redox-coupled ALD of TaO_x is added, and the experimental details of Ta(NMe₂)₅ and Ta(OEt)₅ were shown in **Figure S28**.*

*To understand why the redox coupled ABC-type ALD is favorable to obtaining high selectivity compared with traditional AB-type ALD, various precursors with similar ligands compared with Ta(BⁱBu)(NEt₂)₅ were adopted in **Figure S25-27**, including Nb(NⁱBu)(NEt₂)₃, W(NⁱBu)₂(NMe₂)₂, and Mo(NMe₂)₂(NⁱBu)₂.*

*To figure out the reason for the insignificant mass gain after water pulses using ABC type ALD process, the mass gain as a function of deposition time and temperature on different substrate were added in **Figure S17-18**.*

*To figure out if there are any residual carbon issues from EtOH moieties in TaO_x film, we added the XPS sputter depth measurements of TaO_x film by Ta(NⁱBu)(NEt₂)₃-H₂O and EtOH-Ta(NⁱBu)(NEt₂)₃-H₂O ALD processes in **Figure S21**.*

*To figure out if there is any change in the electrical properties of the TaO_x thin films in applying the reduction-adsorption-oxidation process to TaO_x ALD, we added the C-V tests of both AB-type and ABC-type ALD of TaO_x films in **Figure S22**.*

*The film thickness as a function of the ALD cycles on four substrates at 200 °C by using three ABC-type ALD strategies was added in **Figure S12**.*

*To figure out the surface morphology of QCM crystal oscillator were added in **Figure S19**.*

*In order to verify the influence of three co-reactants on the Cu, the surface of cu substrate was treated with ethanol, water and ozone at 200 °C, which was added in **Figure S8**.*

Reviewer comments, second round

Reviewer #1 (Remarks to the Author):

I recommend publication as is.

Reviewer #2 (Remarks to the Author):

In my opinion, the authors addressed well the concerns raised by the reviewers by providing more details about the performed experiments as well as about the proposed nucleation model. The additional data demonstrating the generality of the proposed ABC growth scheme have been provided.

With this, I think that the revised version of the manuscript (along with the expanded supplementary material) is good for publishing.

Just few very minor remarks:

- Energy units should be added in Table S3
- When describing the set-up for the k-value measurements, please, provide some information about the area of the capacitors.

Reviewer #3 (Remarks to the Author):

The opinions were taken into account and adequately addressed. However, the definition of n^i was omitted in formulas (4-6) of the nucleation modeling section in the supplementary materials. Although it can be inferred from the context that it represents the number of nuclei, providing a precise definition would enhance the reader's comprehension. It is suggested that the manuscript be published after incorporating this clarification, without requiring further review.

Reviewer #2 (Remarks to the Author):

In my opinion, the authors addressed well the concerns raised by the reviewers by providing more details about the performed experiments as well as about the proposed nucleation model. The additional data demonstrating the generality of the proposed ABC growth scheme have been provided. With this, I think that the revised version of the manuscript (along with the expanded supplementary material) is good for publishing. Just few very minor remarks:

- Energy units should be added in Table S3
- When describing the set-up for the k-value measurements, please, provide some information about the area of the capacitors.

Author reply:

We have added energy units “eV” in Table S3 and added details of the C-V tests, such as test frequency, electrode area, etc.

Modifications:

We have added energy units in Table S3 in Supplementary Materials:

Table S3. Calculated energy for different reaction paths on Cu and SiO₂ substrate

Surface		SiO ₂		Cu		
Precursor	Reaction	Eb (eV)	DeltaE (eV)	Reaction	Eb (eV)	DeltaE (eV)
Ta (N ⁱ Bu)(NEt ₂) ₃	pre_ads	0.00	-0.75	pre_ads	0.00	-1.97
	H_trans1	0.37	-0.66	dec1	1.29	-0.76
	HNEt2_des1	0.22	-0.60	dec2	2.26	1.92
Mo(NMe ₂) ₂ (N ⁱ Bu) ₂	pre_ads	0.00	-0.50	pre_ads	0.00	-2.36
	H_trans1	0.33	-0.79	dec1	0.70	0.12
	HNEt2_des2	0.46	0.46	dec2	1.43	0.96
	H_trans1	0.46	-0.46			
	HNEt2_des2	0.80	0.80			
W(N ⁱ Bu) ₂ (NMe ₂) ₂	pre_ads	0.00	-0.39	pre_ads	0.00	-2.15
	H_trans1	0.37	-0.69	dec1	0.79	0.27
	HNEt2_des2	0.00	0.41	dec2	1.56	1.06
	H_trans1	0.59	-0.56			
Nb(N ⁱ Bu)(NEt ₂) ₃	HNEt2_des2	0.74	0.74			
	pre_ads	0.00	-0.97	pre_ads	0.00	-2.01
	H_trans1	0.03	-0.42	dec1	1.36	-0.71
	HNEt2_des1	0.00	-0.07	dec2	2.14	1.82

We have added related discussion in revised manuscript:

“The electrical measurements were performed through capacitance-voltage using a Keithley 4200 impedance analyzer. 100 nm thick Ag film was evaporated as the back electrode, circular Ag electrode with 200 μm diameter and 100 nm thickness was evaporated on target film through a shadow mask served as the front side electrode. Capacitance measurements were conducted at 500 kHz 100 mV ac modulation while the DC gate voltage was swept from -4V to 4V.”

Reviewer #3 (Remarks to the Author):

The opinions were taken into account and adequately addressed. However, the definition of n^i was omitted in formulas (4-6) of the nucleation modeling section in the supplementary materials. Although it can be inferred from the context that it represents the number of nuclei, providing a precise definition would enhance the reader's comprehension. It is suggested that the manuscript be published after incorporating this clarification, without requiring further review.

Author reply:

The definition of n^i in equations (4–6) in the nucleation modeling is added in our supplementary material.

Modifications:

We have added the definition of n^i of the nucleation modeling section in the supplementary materials: “ n^i is the number of nuclei in i^{th} cycle ($n^i = A \cdot N^i$). We re-write Eq. (4) in the following form.”